# MINI-BATCH SUBMODULAR MAXIMIZATION

## ABSTRACT

We present the first *mini-batch* algorithm for maximizing a non-negative monotone *decomposable* submodular function, $F = \sum_{i=1}^{N} f^i$, under a set of constraints. Our experiments demonstrate that a straightforward uniform mini-batch sampling approach significantly outperforms existing state-of-the-art sparsifier methods, requiring only a fraction of their running time. However, explaining this improvement via worst-case analysis is *impossible*.

Instead, we employ *smoothed analysis* to provide a theoretical justification for our empirical findings. Under mild assumptions, we show uniform sampling is superior to weighted sampling for both the mini-batch and sparsifier approaches. We further verify empirically that these assumptions hold across various datasets. Unlike weighted sampling, uniform sampling is simple to implement and several orders of magnitude faster, making it ideal for handling massive real-world datasets.

## 1 INTRODUCTION

Submodular functions capture the natural property of *diminishing returns* which often arises in machine learning, graph theory and economics. For example, imagine you are given a large set of images, and your goal is to extract a small subset that best represents the original set (e.g., creating thumbnails for a YouTube video). Intuitively, this is a submodular optimization problem, as the more thumbnails we have, the less we gain by adding an additional thumbnail. Formally, a set function $F : 2^E \to \mathbb{R}^+$, on a *ground set* $E$, is submodular if for any subsets $S \subseteq T \subseteq E$ and $e \in E \setminus T$, it holds that $F(S + e) - F(S) \geq F(T + e) - F(T)$.

**Decomposable submodular functions** In many natural scenarios $F$ is *decomposable*: $F = \sum_{i=1}^{N} f^i$, where each $f^i : 2^E \to \mathbb{R}^+$ is a non-negative submodular function on the ground set $E$ with $|E| = n$. That is, $F$ can be written as a sum of "simple" submodular functions. We assume that every $f^i$ is represented by an evaluation oracle that, when queried with $S \subseteq E$, returns the value $f^i(S)$. For ease of notation, we assume that $f^i(\emptyset) = 0$ (our results hold even if this is not the case). Our goal is to maximize $F$ under some set of constraints while minimizing the number of oracle calls to $\{f^i\}$. An excellent survey of the importance of decomposable functions is given in (Rafiey & Yoshida, 2022), which we summarize below.

Decomposable submodular functions are prevalent in both machine learning and economics. In economics, they play a pivotal role in welfare optimization during combinatorial auctions (Dobzinski & Schapira, 2006; Feige, 2009; Feige & Vondrák, 2006; Papadimitriou et al., 2008; Vondrák, 2008). In machine learning, they are instrumental in tasks like data summarization, aiming to select a concise yet representative subset of elements. Their utility spans various domains, from exemplar-based clustering by (Dueck & Frey, 2007) to image summarization (Tschiatschek et al., 2014), recommender systems (Parambath et al., 2016) and document summarization (Lin & Bilmes, 2011). The optimization of these functions, especially under specific constraints (e.g., cardinality, matroid), has been studied in various data summarization settings (Mirzasoleiman et al., 2016a;b;c) and differential privacy (Chaturvedi et al., 2021; Mitrovic et al., 2017; Rafiey & Yoshida, 2020). In many of the above applications, $N$ (the number of underlying submodular functions) is extremely large, making the evaluation of $F$ prohibitively slow. We illustrate this with a simple example.

**Example (Welfare maximization)** Imagine you are tasked with deciding on a meal menu for a large group of $N$ people (e.g., all students in a university, all high school students in a country). You need to choose $k$ ingredients to use from a predetermined set (chicken, fish, beef, etc.) of size $n$. Every student has a specific preference, modeled as a monotone submodular function $f^i$. Our goal

is to maximize the "social welfare", $F(S) = \sum f^i(S)$, over all students. We use this meal menu example for simplicity and ease of presentation, but the same framework applies to more meaningful domains such as public service allocation, medical benefit selection, or emergency response planning.

**The greedy algorithm** Going forward we focus on the case where $\forall i, f^i$ is monotone ($\forall S \subseteq T \subseteq E, f^i(T) \geq f^i(S)$), which in turn means that $F$ is monotone. This is applicable to the above example (e.g., students are happier with more food varieties). For ease of presentation, let us first focus on maximizing $F$ under a cardinality constraint $k$, i.e., $\max F(S), |S| \leq k$. The classical greedy algorithm (Nemhauser et al., 1978) (presented below) achieves an *optimal* $(1 - 1/e)$-approximation for this problem. Let us define some notation before we continue. For $S, A \subseteq E$ we define $F_S(A) = F(S + A) - F(S)$. We slightly abuse notation and write $F_S(e), F(e)$ instead of $F_S(\{e\}), F(\{e\})$.

**Greedy:** Start by initializing an empty set $S_1$. Then for each $j$ from 1 to $k$, do the following: Find the element $e' \in E \setminus S_j$ that maximizes the function $F_{S_j}$ and add it to $S_j$ to form the new set $S_{j+1}$. After completing all iterations, return $S_{k+1}$.

When $F$ is decomposable, and each evaluation of $f^i$ is counted as an oracle call, the above algorithm requires $O(Nnk)$ oracle calls. This can be prohibitively expensive if $N \gg n$. Looking back at our example, Greedy will require asking every student $k$ questions. That is, we would start by asking all students: "Rate how much you would like to see food $x$ on the menu" for $n$ possible options. We would then need to wait for all of their replies, and continue with "Given that chicken is on the menu, rate how much you would like to see food $y$ on the menu" for $n - 1$ possible options and so on. Even if we do an online poll, this is still very time-consuming (as we must wait for *everyone* to reply in each of the $k$ steps). If we could *sample* a representative subset of the students, we could greatly speed up the above process. Specifically, we would like to eliminate the dependence on $N$.

**A sparsifier-based approach** Let $\hat{F} = \sum_{i=1}^{N} w_i f^i$ where $w \in \mathbb{R}^N$. We say that $\hat{F}$ is an $\epsilon$-sparsifier (Rafiey & Yoshida, 2022) for $F$ if it satisfies with high probability (w.h.p)[1] that $\forall S \subseteq E, (1 - \epsilon)F(S) \leq \hat{F}(S) \leq (1 + \epsilon)F(S)$. The size of the sparsifier is the number of non-zero entries in $w$. The current state-of-the-art construction (Kenneth & Krauthgamer, 2023) sets $w_i = 1/\alpha_i$ with probability $\alpha_i = \min\{p_i\alpha, 1\}$ where $p_i = \max_{e \in E, F(e) \neq 0} \frac{f^i(e)}{F(e)}$ and $\alpha$ is a parameter to control the size of the sparsifier. Assuming the size of the solution is bounded by $k$, it is possible to construct a sparsifier of size $\widetilde{O}(k^2 n \epsilon^{-2})$ using $O(Nn)$ oracle calls[2]. Note that, when the size of the solution is bounded by $k$ it is enough for the sparsifier guarantees to only hold for sets of size at most $k$.

Computing the $p_i$'s is the main bottleneck of this approach, and is treated as a *preprocessing* step in prior work. Nevertheless, it still contributes to the overall query complexity of the algorithm. We count the preprocessing step toward the overall query complexity (i.e., we include all sparsifier-construction queries in the total number of queries performed by the algorithm).

The simplest way to overcome the $O(Nn)$ bottleneck in the sparsifier construction is to use *uniform sampling* (i.e., set $\forall i \in [N], p_i = 1/N$). We experimentally evaluate the uniform sparsifier against the weighted sparsifier and observe that it achieves solutions of equal (or superior) quality while reducing the query complexity by orders of magnitude. Surprisingly, this simple baseline was neglected in previous experimental evaluations (Rafiey & Yoshida, 2022).

Unfortunately, we cannot get worst-case theoretical guarantees for uniform sampling. Consider the case where only a single $f^i$ takes non-zero values (all other $f^i$'s are always 0). Clearly, uniform sampling will almost surely miss $f^i$. However, this is a pathological case that is very unlikely to occur in practice. To bridge this gap, we go beyond worst-case analysis and consider the *smoothed complexity* of this problem.

BEYOND WORST-CASE ANALYSIS

Smoothed analysis was introduced by (Spielman & Teng, 2004) in an attempt to explain the fast runtime of the Simplex algorithm in practice, despite its exponential worst-case runtime. In the

---

[1]Probability at least $1 - 1/n^c$ for an arbitrary constant $c > 1$. The value of $c$ does not affect the asymptotics of the results we state (including our own).

[2]Where $\widetilde{O}$ hides $\log n$ factors.

smoothed analysis framework, an algorithm is provided with an adversarial input that is perturbed by some random noise. Often, the crux of smoothed analysis is defining a realistic model of noise.

Our main theoretical contribution is defining a very natural smoothing model. To define our model, we assume, without loss of generality, that $\forall i \in [N], e \in E, f^i(e) \in [0, 1]$ (we can always achieve this by normalization). Let $\phi \in [0, 1]$ be a *smoothing* parameter and $d \in [N]$ be a *dependency* parameter.

**Smoothing model**     We assume that $N = \Omega(\frac{d}{\phi} \log(nd))$, and there *exists* an element, $e^* \in E$, such that the following two conditions hold: (1) Every $f^i(e^*)$ is a random variable such that $\mathbb{E}[f^i(e^*)] \geq \phi$. (2) Within $\left\{ f^i(e^*) \right\}_{i \in [N]}$ every random variable depends on at most $d$ others. These assumptions are very mild, as we assume that for every $e \neq e^*$ the values $\left\{ f^i(e) \right\}_{i \in [N]}$ may be set *adversarially*.

**Intuition**     Going back to our lunch menu example, Condition (1) means that there *exists* a food on the menu that is not universally hated by the students. Condition (2) means that the preference of a student for that specific food is sufficiently independent of the preferences of other students. Assuming that $N = \Omega(\frac{d}{\phi} \log(nd))$ means that we have a sufficiently large student body – note that increasing $N$ should not change $\phi$ and $d$.

**Comparison to other models of smoothing**     Perhaps the most general smoothing approach when dealing with weighted inputs (e.g., in $[0, 1]$) is to assume that the weights are taken from some distribution whose density is upper bounded by a smoothing parameter $\phi$ (Etscheid & Röglin, 2017; Angel et al., 2017). This generalizes the approach of (Spielman & Teng, 2004), where Gaussian noise was added to the weights. Our approach is even more general, as the above immediately implies that the expectation is lower bounded by $\phi$. Furthermore, we only assume bounded dependency and only *partially* smooth the input.

**A parametrized perspective**     All of our theoretical results only require that $\max_{e \in E} F(e) \geq \phi N$ – i.e., that the largest singleton gain is a $\phi$-approximation to the optimum. We could treat $\max_{e \in E} F(e)$ as a parameter of $F$, similar to curvature[3]. However, the smoothing model provides more insight into *why* this property holds. We believe that it is a realistic assumption that each $f^i$ is drawn from some distribution, and that for at least one $e^* \in E$ this distribution meets our mean-and-dependency conditions, which in turn guarantees the required singleton-gain bound.

CONTRIBUTION AND TECHNIQUES

Our main contribution is introducing the mini-batch greedy algorithm with uniform sampling (Algorithm 2 in Section 2). This is a natural step, as mini-batch algorithms are widely used in practice (e.g., SGD, mini-batch $k$-means (Sculley, 2010)). Instead of sampling a single sparsifier and executing the algorithm, we sample a new sparsifier (mini-batch) in every iteration. Our algorithm can be seen as an analogue to Sculley's mini-batch $k$-means algorithm (Sculley, 2010), a simple and highly practical method described in a concise two-page paper, which has since been widely adopted in libraries such as scikit-learn (Pedregosa et al., 2011).

We experimentally observe that the mini-batch algorithm achieves solutions of superior quality to the sparsifier-based approach. We provide theoretical guarantees for both our mini-batch algorithm and the uniform sparsifier under our smoothing model.

**Theoretical results**     We start our theoretical analysis (Section 2) by presenting a meta greedy algorithm (Algorithm 1) which captures the behavior of the greedy algorithm under various constraints. We focus on cardinality and $p$-system constraints. The cardinality constraint was chosen for its simplicity and applicability, while the $p$-system constraint was chosen for its generality.

It is known that for cardinality and $p$-system constraints, this algorithm achieves near-optimal approximation guarantees even if we only have a multiplicative approximation to $F_{S_j}$ (the marginal gain given the current partial solution) in every iteration. We generalize these guarantees to hold under an *additive approximation*.

Finally, in Section 3 we show that, under our smoothing model, the mini-batch algorithm with uniform sampling indeed achieves the desired additive approximation to $F_{S_j}$ in every iteration. Compared

---

[3]The *curvature* is defined as $c = 1 - \min_{S \subseteq E, e \in E \setminus S} \frac{F_S(e)}{F(e)}$. We say that $F$ has *bounded-curvature* if $c < 1$.

to weighted sampling, we eliminate an $O(n)$-multiplicative factor in the query complexity, while incurring a multiplicative $O(1/\phi)$ factor. This gives a smooth transition in the query complexity between the worst-case scenario where $\phi \to 0$ and the well-behaved case where $\phi = \Theta(1)$.

We use similar techniques to show that under our smoothing assumptions the uniform sparsifier achieves the same query complexity as our mini-batch algorithm. Furthermore, we introduce a *weighted* mini-batch algorithm and show that it achieves improved *worst-case* guarantees compared to the weighted sparsifier. Our weighted algorithm uses the sampling probabilities of (Kenneth & Krauthgamer, 2023), which brings back the $O(Nn)$ bottleneck to the query complexity, making the algorithm impractical. We include it primarily for theoretical completeness.

In the main body of the paper we focus only on cardinality constraints and the uniform mini-batch algorithm for ease of presentation. Nevertheless, this is enough to present our main ideas and contributions. We include a discussion of $p$-systems in Appendix B, our results for the uniform sparsifier in Appendix E and for the weighted mini-batch algorithm in Appendix F.

We compare our results with the state-of-the-art sparsifier results and the naive algorithm (without sampling or sparsification) in Table 1. **Our main results are the smoothed complexity guarantees for uniform sampling which explain why it outperforms weighted sampling in practice**.

| | Oracle queries | |
| --- | --- | --- |
| | Weighted (Worst-case) | Uniform (Smoothed) |
| Naive | $O(Nnk)$ (Nemhauser et al., 1978) | |
| Sparsifier | $\widetilde{O}\big(Nn + \frac{k^3 n^2}{\epsilon^2}\big)$ (Kenneth & Krauthgamer, 2023) | $\widetilde{O}\big(\frac{k^2 n}{\epsilon^2 \phi}\big)$ [this work] |
| Mini-batch | Card. $\widetilde{O}\big(Nn + \frac{k^2 n^2}{\epsilon^2}\big)$ [this work] 
 $p$-sys. $\widetilde{O}\big(Nn + \frac{k^2 p n^2}{\epsilon^2}\big)$ [this work] | |
| Mini-batch (bounded curvature) | $\widetilde{O}\big(Nn + \frac{kn^2}{(1-c)\epsilon^2}\big)$ [this work] | - |

Table 1: Comparison of the expected number of oracle queries for the greedy algorithm. The results for the uniform column hold in expectation and w.h.p. Results are under both a cardinality constraint and a $p$-system constraint. Unless explicitly stated, the number of queries is the same for both constraints. All results achieve the near-optimal approximation guarantees of $(1 - 1/e - \epsilon)$ for a cardinality constraint and $\big(\frac{1-\epsilon}{p+1}\big)$ for a $p$-system constraint.

**Experimental evaluation** In Section 4 we compare the uniform mini-batch algorithm against the weighted mini-batch algorithm, the (weighted) sparsifier of (Kenneth & Krauthgamer, 2023) and the uniform sparsifier on five datasets: CIFAR100, FashionMNIST, Uber pickups, Discogs, Facebook. We evaluate our algorithm both under a cardinality and a $p$-system constraint.

We observe a considerable performance discrepancy in favor of the uniform mini-batch algorithm for small batch sizes. The weighted methods are orders of magnitude slower due to the $O(Nn)$ factor. We observe similar results when we combine our approach with the popular *stochastic-greedy* algorithm (Buchbinder et al., 2015; Mirzasoleiman et al., 2015).

**Empirical validation of the smoothing model** In addition to evaluating our algorithm, we empirically examine the validity of our smoothing model[4]. Specifically, we estimate $\phi$ by computing $\min_e \frac{1}{N} \sum_i f^i(e)$. The empirical values are as follows: CIFAR100: 0.38, FashionMNIST: 0.35, Uber Pickup: 0.61, Discogs: 0.13, Facebook: 0.04. This suggests that $\phi = \Theta(1)$ in realistic settings.

We then ask a natural follow-up: could real-world datasets support a stronger smoothing model? In particular, we consider a variant in which the assumptions hold for *all* $e \in E$, rather than for just a single $e^*$. To evaluate this, we compute $\max_e \frac{1}{N} \sum_i f^i(e)$, and observe much smaller values: CIFAR100: 0.34, FashionMNIST: 0.31, Uber Pickup: $2.36 \times 10^{-4}$, Discogs: $4.10 \times 10^{-6}$, Facebook: $2.5 \times 10^{-4}$. For three of the five datasets, the empirical $\phi$ is essentially zero, indicating that such a

---

[4]Strictly speaking, what we confirm is a lower bound on the largest singleton gain, which is all that is required for our theoretical guarantees to hold.

strong smoothing assumption is unrealistic. These results further justify the weak assumptions of our smoothing model.

## 2 THE MINI-BATCH ALGORITHM

**Meta greedy algorithm**   Our starting point is the meta greedy algorithm (Algorithm 1). The algorithm executes for $k \leq n$ iterations, where $k$ is some upper bound on the size of the solution. At every iteration, the set $A_j \subseteq E \setminus S_j$ represents some constraint that limits the choice of potential elements to extend $S_j$. The algorithm terminates when the solution size reaches $k$ or when no further extensions to the current solution are possible (i.e., $A_j = \emptyset$). Furthermore, the algorithm does not have access to the exact *incremental oracle*, $F_{S_j}$, at every iteration, but only to some approximation (which may differ between iterations).   Before we formally define "approximation", let us note

---

**Algorithm 1:** Meta greedy algorithm with an approximate oracle

1  $S_1 \leftarrow \emptyset$
2  Let $k$ be an upper bound on the size of the solution
3  **for** $j = 1$ *to* $k$ **do**
4  $\quad$ Let $A_j \subseteq E \setminus S_j$   ▷ Problem specific constraint (e.g., $A_j = E \setminus S_j$ for card. constraint)
5  $\quad$ **if** $A_j = \emptyset$ **then** return $S_j$
6  $\quad$ Let $\hat{F}_{S_j}^j$ be an approximation for $F_{S_j}$   ▷ Problem specific approximation
7  $\quad$ $e_j = \arg\max_{e \in A_j} \hat{F}_{S_j}^j(e)$
8  $\quad$ $S_{j+1} = S_j + e_j$
9  **end**
10  return $S_{k+1}$

---

that when we have access to exact values of $F_{S_j}$, Algorithm 1 captures many variants of the greedy submodular maximization algorithm. For example, setting $A_j = E \setminus S_j$ we get `Greedy`. This meta-algorithm also captures the case of maximization under a *$p$-system constraint*. For ease of presentation, we defer the discussion about $p$-systems to Appendix B.

**Approximate oracles**   In many scenarios we do not have access to *exact* values of $F_{S_j}$, and instead we must make do with an approximation. We start with the notion of an *approximate incremental oracle* introduced in (Goundan & Schulz, 2007). We say that $\hat{F}_{S_j}^j$ is an $(1 - \epsilon)$-approximate incremental oracle if $\forall e \in A_j, (1 - \epsilon)F_{S_j}(e) \leq \hat{F}_{S_j}^j(e) \leq (1 + \epsilon)F_{S_j}(e)$. It was shown in (Goundan & Schulz, 2007; Călinescu et al., 2011)[5] that given a $(1 - \epsilon)$-approximate incremental oracle, the greedy algorithm under both a cardinality constraint and a $p$-system constraint achieves almost the same (optimal) approximation ratio as with an exact oracle.

**Theorem 2.1.** *Algorithm 1 with a $(1 - \epsilon)$-approximate incremental oracle has the following guarantees: (1) It achieves a $(1 - 1/e - \epsilon)$-approximation under a cardinality constraint $k$. (2) It achieves a $\left(\frac{1-\epsilon}{1+p}\right)$-approximation under a $p$-system constraint.*

We introduce a weaker type of approximate incremental oracle, which we call an *additive* approximate incremental oracle. We extend the results of Theorem 2.1 for this case. Let $S^*$ be some optimal solution for $F$ (under the relevant set of constraints). We say that $\hat{F}_{S_j}^j$ is an *additive* $\epsilon'$-approximate incremental oracle if $\forall e \in A_j, F_{S_j}(e) - \epsilon' F(S^*) \leq \hat{F}_{S_j}^j(e) \leq F_{S_j}(e) + \epsilon' F(S^*)$.

This might seem problematic at first glance, as it might be the case that $F(S^*) \gg F_{S_j}(e)$. Luckily, the proofs guaranteeing the approximation ratio are *linear*. Therefore, by the end of the proof, we end up with an expression of the form: $F(S_{k+1}) \geq F(S^*)\beta - \gamma\epsilon' F(S^*)$. Where $\beta$ is the desired approximation ratio and $\gamma$ depends on the constraints (e.g., $\beta = (1 - 1/e), \gamma = 2k$ for a cardinality constraint). We can achieve the desired result by setting $\epsilon' = \epsilon/\gamma$. We state the following theorem

---

[5]Strictly speaking, both (Goundan & Schulz, 2007; Călinescu et al., 2011) define the approximate incremental oracle to be a function that returns $e_j$ at iteration $j$ of the greedy algorithm such that $\forall e \in A_j, F_{S_j}(e_j) \geq (1 - \epsilon)F_{S_j}(e)$. Our definition guarantees this property while allowing easy analysis of the mini-batch algorithm.

(the proofs are similar to those of (Goundan & Schulz, 2007; Călinescu et al., 2011), and we defer them to the Appendix).

**Theorem 2.2.** *Algorithm 1 with an additive $\epsilon'$-approximate incremental has the following guarantees: (1) If $\epsilon' < \epsilon/2k$, it achieves a $(1 - 1/e - \epsilon)$-approximation under a cardinality constraint $k$. (2) If $\epsilon' < \epsilon/2kp$, it achieves a $(\frac{1-\epsilon}{1+p})$-approximation under a $p$-system constraint.*

**Mini-batch sampling**  We present our mini-batch algorithm in Algorithm 2. Here, the approximate incremental oracle is constructed via mini-batch sampling with a batch size parameter $\alpha$. In Section 3 we analyze the relation between $\alpha$ and the type of approximate incremental oracle guaranteed by our sampling procedure. We state the main theorem for Section 3 below (recall that $e^*$ is the element guaranteed by the smoothing assumptions and $S^*$ is an optimal solution for $F$).

**Theorem 2.3.** *Under the smoothing assumptions, when applied to maximize a non-negative monotone submodular function with batch size $\alpha = \Theta\left(\epsilon^{-2}\gamma \log n\right)$, the mini-batch greedy algorithm (Algorithm 2) guarantees that w.h.p for every $j \in [k]$, $\hat{F}_{S_j}^j$ is an additive $(\epsilon F(e^*)/\gamma F(S^*))$-approximate incremental oracle, for any $\gamma > 0$. Furthermore, the algorithm performs $O(\alpha kn)$ oracle queries in expectation and w.h.p.*

---

**Algorithm 2:** Mini-batch greedy

1   $\alpha$ is a batch size parameter (where $\alpha \le N$), $k$ is an upper bound on the size of the solution
2   $S_1 \leftarrow \emptyset$
3   **for** $j = 1$ *to* $k$ **do**
4      Let $A_j \subseteq E \setminus S_j$    ▷ Problem specific constraint
5      **if** $A_j = \emptyset$ **then** return $S_j$
6      $w \leftarrow \vec{0}$
7      $\forall i, w_i \leftarrow N/\alpha$ with probability $\alpha/N$
8      $\hat{F}^j = \sum_{i=1}^N w_i f^i$
9      $e_j = \arg\max_{e \in A_j} \hat{F}_{S_j}^j(e)$
10      $S_{j+1} = S_j + e_j$
11   **end**
12   return $S_{k+1}$

---

Finally, we state our main result for this section.

**Theorem 2.4.** *Under the smoothing assumptions, when applied to maximize a non-negative monotone submodular function the mini-batch greedy algorithm (Algorithm 2) achieves w.h.p a $(1 - 1/e - \epsilon)$-approximation under a cardinality constraint and a $(\frac{1-\epsilon}{1+p})$-approximation under a $p$-system constraint using $O(\frac{k^2 n \log n}{\epsilon^2 \phi})$ oracle evaluations in expectation and w.h.p.*

*Proof.* Combining Theorem 2.3 (setting $\gamma = 2k$) with Theorem 2.2 we get our result for cardinality constraints. For $p$-systems, we break the proof into cases. If $F(e^*) \ge F(S^*)/p$, Theorem 2.3 with $\gamma > 1$ guarantees that $\hat{F}_{S_1}^1(e^*) = \hat{F}^1(e^*) \ge F(e^*) - \epsilon F(e^*) = (1 - \epsilon)F(e^*)$, which guarantees that $e_1$, the element chosen in the first iteration of the greedy algorithm, is a $(1 - \epsilon)/p$ approximate solution for $F$, and the rest follows because $F$ is monotone. If $F(e^*) < F(S^*)/p$, Theorem 2.3 guarantees an additive $(\epsilon/\gamma p)$-approximate oracle. Setting $\gamma = 2k$ completes the proof. ☐

## 2.1 RELATED WORK

**Approximate oracles**  Apart from the results of (Goundan & Schulz, 2007; Călinescu et al., 2011) some works use different notions of an approximate oracle. Several works consider an approximate oracle $\hat{F}$, such that $\forall S \subseteq E, \left|\hat{F}(S) - F(S)\right| < \epsilon F(S)$ (Crawford et al., 2019; Horel & Singer, 2016; Qian et al., 2017). The main difference of these models from our work is that they do not assume the surrogate function, $\hat{F}$, to be submodular. This adds a significant complication to the analysis and degrades the performance guarantees.

**Mini-batch methods**  The closest result resembling our mini-batch approach is the *stochastic-greedy algorithm* (Buchbinder et al., 2015; Mirzasoleiman et al., 2015). They improve the expected

query complexity of the greedy algorithm under a cardinality constraint by only considering a small random subset of $E \setminus S_j$ at the $j$-th iteration. We note that their approach can be combined into our mini-batch algorithm, reducing our query complexity by a $\tilde{\Theta}(k)$ factor, resulting in an approximation guarantee in expectation instead of w.h.p.

**Smoothed analysis** To the best of our knowledge, (Rubinstein & Zhao, 2022) is the only result that considers smoothed analysis in the context of submodular maximization. They consider submodular maximization under a cardinality constraint, where the cardinality parameter $k$ undergoes a perturbation according to some known distribution.

## 3  ANALYSIS OF THE MINI-BATCH GREEDY ALGORITHM

We start by stating the following useful Chernoff bounds:

**Theorem 3.1** (Chernoff bound (Motwani & Raghavan, 1995)). *Let $X_1, ..., X_N$ be independent random variables in the range $[0, a]$. Let $X = \sum_{i=1}^{N} X_i$. Then for any $\epsilon \in [0, 1]$ and $\mu \geq \mathbb{E}[T]$ it holds that $\mathbb{P}(|X - \mathbb{E}[X]| \geq \epsilon\mu) \leq 2\exp\left(-\epsilon^2\mu/3a\right)$.*

**Theorem 3.2** (Bounded dependency Chernoff bound (Pemmaraju, 2001)). *Let $X_1, ..., X_N$ be identically distributed random variables in the range $[0, 1]$ with bounded dependency $d$. Let $X = \sum_{i=1}^{N} X_i$. Then for any $\epsilon \in [0, 1]$ and $\mu = \mathbb{E}[T]$ it holds that $\mathbb{P}(|X - \mathbb{E}[X]| \geq \epsilon\mu) \leq \Theta(d) \cdot \exp\left(-\Theta(\epsilon^2\mu/d)\right)$.*

Let us first lower bound $F(e^*)$ using the smoothing assumptions.

**Lemma 3.3.** *Under the smoothing assumptions it holds w.h.p that $F(e^*) \geq N\phi/2$.*

*Proof.* According to the smoothing assumptions, it holds that $\mathbb{E}[F(e^*)] = \sum_{i \in [N]} \mathbb{E}[f^i(e^*)] \geq N\phi$. Applying the bounded dependency Chernoff bound with $\epsilon = 1/2, \mu = \mathbb{E}[F(e^*)]$ we get that:

$$\mathbb{P}\left[|F(e^*) - \mathbb{E}[F(e^*)]| \geq \mu/2\right] \leq \Theta(d)\exp(-\Theta(\mu/d)) \leq \Theta(d)\exp(-\Theta(N\phi/d)) \leq 1/n^2$$

Where the last inequality is due to the fact that $N = \Omega((d/\phi)\log(nd))$. We conclude that w.h.p $F(e^*) \geq N\phi/2$. □

Going forward, we condition on the high probability event in Lemma 3.3 and no longer treat any $f^i$ as a random variable. This is justified because all $f^i$'s are determined prior to the execution of the algorithm.

The following lemma provides concentration guarantees for $\hat{F}^j$ in Algorithm 2.

**Lemma 3.4.** *For every $j \in [k]$ and $S \subseteq E$ ($\hat{F}^j$ sampled after $S$ is fixed) and for every $e \in E$ and $\mu \geq F_S(e)$, it holds that $\mathbb{P}\left[|\hat{F}_S^j(e) - F_S(e)| \geq \epsilon\mu\right] \leq 2\exp\left(-\frac{\epsilon^2\alpha\mu}{6F(e^*)}\right)$.*

*Proof.* Fix some $e \in E$. Due to the fact that $\forall i \in [N], \mathbb{E}[w_i] = 1$ we have $\mathbb{E}[\hat{F}_S^j(e)] = \mathbb{E}[\sum_{i \in [N]} w_i f_S^i(e)] = F_S(e)$. Applying a Chernoff bound (Theorem 3.1) we have $\mathbb{P}\left[|\hat{F}_S^j(e) - F_S(e)| \geq \epsilon\mu\right] \leq 2\left(-\frac{\epsilon^2\mu}{3a}\right)$ where $a = \max\{\frac{N}{\alpha}f_S^i(e)\}_{i \in [N]} \leq N/\alpha$ (recall that all $f^i$'s are upper bounded by 1). Let us upper bound $a$ as a function of $F(e^*)$ where $e^* \in E$ is the element guaranteed by the smoothing assumption. Rearranging Lemma 3.3 we get that $N \leq 2F(e^*)/\phi$. We conclude that $a \leq N/\alpha \leq 2F(e^*)/\phi\alpha$. Given the above upper bound for $a$ we get: $\mathbb{P}\left[|\hat{F}_S^j(e) - F_S(e)| \geq \epsilon\mu\right] \leq 2\left(-\frac{\epsilon^2\mu}{3a}\right) \leq 2\exp\left(-\frac{\epsilon^2\alpha\mu}{6F(e^*)}\right)$. □

We are now ready to prove the main theorem for this section.

**Theorem 2.3.** *Under the smoothing assumptions, when applied to maximize a non-negative monotone submodular function with batch size $\alpha = \Theta\left(\epsilon^{-2}\gamma\log n\right)$, the mini-batch greedy algorithm (Algorithm 2) guarantees that w.h.p for every $j \in [k]$, $\hat{F}_{S_j}^j$ is an additive $(\epsilon F(e^*)/\gamma F(S^*))$-approximate incremental oracle, for any $\gamma > 0$. Furthermore, the algorithm performs $O(\alpha kn)$ oracle queries in expectation and w.h.p.*

*Proof.* The number of oracle evaluations is due to the fact that every $w_i$ is taken to be non-zero with probability $\alpha/N$, which results in an expected $\alpha$ non-zero entries per iteration. Due to the size of $\alpha$ we also get concentration via a Chernoff bound. The algorithm executes for $k$ iterations and must evaluate $|A_j| \leq n$ elements per iteration.

Let us prove the approximation guarantees. We break the analysis into cases.

$F(e^*) \leq \gamma F_{S_j}(e)$**:**    Setting $\mu = F_{S_j}(e)$ we get:

$$\mathbb{P}\left[|\hat{F}_{S_j}^j(e) - F_{S_j}(e)| \geq \epsilon F_{S_j}(e)\right] \leq 2\exp\left(-\frac{\epsilon^2 \alpha F_{S_j}(e)}{6F(e^*)}\right) \leq 2\exp\left(-\frac{\epsilon^2 \alpha}{6\gamma}\right) \leq 1/n^3$$

$F(e^*) > \gamma F_{S_j}(e)$**:**    Here we can set $\mu = F(e^*)/\gamma \geq F_{S_j}(e)$ and get:

$$\mathbb{P}\left[|\hat{F}_{S_j}^j(e) - F_{S_j}(e)| \geq \epsilon F(e^*)/\gamma\right] \leq 2\exp\left(-\frac{\epsilon^2 \alpha F(e^*)}{6\gamma F(e^*)}\right) \leq 2\exp\left(-\frac{\epsilon^2 \alpha}{6\gamma}\right) \leq 1/n^3$$

In both cases, the last inequality is by setting an appropriate constant in $\alpha = \Theta(\epsilon^{-2}\gamma \log n)$. Note that in the first case we get a $(1 - \epsilon)$-approximate incremental oracle and in the second an additive $(\epsilon/\gamma)$-approximate incremental oracle. The second case is the worst of the two for our analysis. We take a union bound over all $e \in E$ and $j \in [k]$ (at most $n^2$ values), which concludes the proof. □

## 4   EXPERIMENTS

We perform experiments on the following datasets, where the goal for all datasets is to maximize $F(S)$ subject to $|S| \leq k$. For influence maximization, we also enforce a $p$-system constraint.

**Uber pickups**    This dataset consists of Uber pickups in New York City in May 2014[6]. The set contains $652,434$ records, where each record consists of a longitude and latitude, representing a pickup location. Following (Rafiey & Yoshida, 2022) we aim to find $k$ positions for idle drivers to wait from a subset of popular pickup locations. We formalize the problem as follows. We run Lloyd's algorithm on the dataset, $X$, and find 100 cluster centers. We set these cluster centers to be the ground set $E$. We define the goal function $F : 2^E \to \mathbb{R}^+$ as $F(S) = \sum_{v \in X} f_v(S)$ where $f_v(S) = \max_{e \in E} d(v, e) - \min_{e \in S} d(v, e)$, and $d(v, e)$ is the Manhattan distance between $v$ and $e$.

**Discogs (Kunegis, 2013)**    This dataset[7] provides audio record information structured as a bipartite graph $G = (L, R; E')$. The left nodes represent labels, and the right nodes represent styles. Each edge $(u, v) \in L \times R$ signifies the involvement of a label $u$ in producing a release of a style $v$. The dataset comprises $|R| = 383$ labels, $|L| = 243,764$ styles, and $|E'| = 5,255,950$ edges. We aim to select $k$ styles that cover the activity of the maximum number of labels. We construct a maximum coverage function $F : 2^R \to \mathbb{R}$, where $F(S) = \sum_{v \in L} f_v(S)$, and $f_v(S)$ equals 1 if $v$ is adjacent to some element of $S$ and 0 otherwise.

**Exemplar-based clustering**    We consider the problem of selecting a representative subset of $k$ images from a massive data set. We present experiments for both the CIFAR100 and the FashionMNIST datasets. For each dataset, we consider a subset of $50,000$ images, denoted by $X$. We flatten every image into a one-dimensional vector, subtract from it the mean of all images, and normalize it to unit norm. We take the distance between two elements in $X$ as $d(x, x') = \|x - x'\|^2$. Here, the ground set is simply the dataset, $E = X$. Similarly to the Uber pickup dataset we define the goal function $F : 2^E \to \mathbb{R}^+$ as $F(S) = \sum_{v \in X} f_v(S)$ where $f_v(S) = \max_{e \in E} d(v, e) - \min_{e \in S} d(v, e)$.

**Influence Maximization**    We consider the problem of maximizing influence in a social network. We use the Facebook dataset from SNAP (McAuley & Leskovec, 2012)[8], a simple undirected graph of real user connections with $4,039$ nodes and $88,234$ edges. We study influence spread under the Independent Cascade Model (Kempe et al., 2015): each activated node has one chance to activate each neighbor with probability $p$ (here $p = 0.01$). The goal is to select a seed set $S \subseteq V$ ($V$ is the ground set) of size $k$ to maximize the expected total number of activated nodes. Furthermore, we

---

[6]https://www.kaggle.com/fivethirtyeight/uber-pickups-in-new-york-city
[7]http://konect.cc/networks/discogs_lstyle/
[8]https://snap.stanford.edu/data/ego-Facebook.html

enforce an independence constraint - the nodes in $S$ may not be adjacent. This corresponds to a $p$-system, where $p$ is the maximum degree in the graph. Formally, let $F(S) = \sum_{v \in V} f_v(S)$ be the expected cascade size where $f_v(S)$ is the expectation that $v$ is activated by $S$ - we approximate this via 500 Monte Carlo runs.

**Experimental setup**    We compare the sparsifier and mini-batch approaches on each of the above datasets. For both, we evaluate the weighted and uniform variants. We introduce a parameter $\beta \in (0, 1)$ which corresponds to the desired fraction of *data* (the elements that constitute $F$) we wish to sample. We set $\alpha$ for all algorithms such that the expected number of sampled elements is $\beta N$ in the mini-batch / sparsifier. For example, setting $\beta = 0.01$ means that the sparsifier will sample 1% of the data and execute the greedy algorithm, while the mini-batch algorithm will sample 1% of the data for every iteration of the greedy algorithm.

In practice, the naive algorithm is usually augmented with the following heuristic called *lazy-greedy* (Minoux, 2005). The algorithm initially creates a max heap of $E$ ordered according to a key $\rho(e)$, where initially $\rho(e) = F(e)$. During the $j$-th iteration, it repeatedly pops $e$ from the top of the heap, updates its key $\rho(e) = F_{S_j}(e)$, and inserts it back into the heap. Due to submodularity, if $e$ remains at the top of the heap after the update, we know it maximizes $F_{S_j}(e)$. While this does not change the asymptotic number of oracle queries, it often leads to significant improvements in practice.

We also consider *stochastic-greedy* (Buchbinder et al., 2015; Mirzasoleiman et al., 2015), which further reduces the number of oracle evaluations by greedily choosing the next element added to $S_j$ from a subset of $E \setminus S_j$ of size $\frac{n}{k} \log \frac{1}{\epsilon}$ sampled uniformly at random. We apply both of the above to the sparsifier / mini-batch approach and compare them against lazy-greedy and stochastic-greedy.

In Figure 1 we plot results for different $\beta$ values for both the sparsifier and the mini-batch approach for different values of $k$. Every point on the graph is the average of 20 executions with the same $k, \beta$. For every dataset, we present two plots: (a) the relative utility (value of $F$) of the mini-batch / sparsifier approach compared to lazy-greedy, (b) the relative number of oracle evaluations. To allow different $\beta$ values to fit on the same plot, we use a logarithmic scale in (b). We use "u" / "w" prefixes for uniform / weighted sampling. In Figures 3,4 (Appendix D) we compare the sparsifier and mini-batch algorithms (augmented with the sampling scheme of stochastic-greedy) to stochastic-greedy.

**Results**    The first clear observation is that weighted sampling is impractical due to its $O(Nn)$ cost bottleneck. In the cost plots, the curves for all $\beta$ values collapse into a single line. Surprisingly, the relative performance of uniform sampling is overall better than weighted sampling. Finally, we observe that the mini-batch algorithm is superior to the sparsifier approach for small values of $\beta$ while using about the same number of queries.

The largest performance gap occurs on the Facebook dataset. This is partly due to the independence constraint, which complicates cost normalization across methods. Still, we observe that the uniform mini-batch method outperforms the uniform sparsifier for $\beta = 10^{-1}$, where the number of queries is about the same, and even for $\beta = 10^{-3}$, where the sparsifier uses considerably more queries.

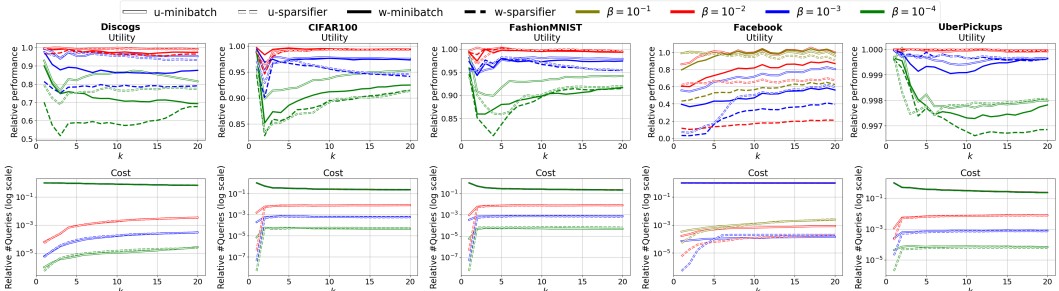

Figure 1: Sparsifier and mini-batch compared using lazy-greedy. We present plots for $\beta \in \{10^{-2}, 10^{-3}, 10^{-4}\}$ for all datasets except Facebook where we plot $\beta \in \{10^{-1}, 10^{-2}, 10^{-3}\}$ due to its small size. A larger version of this figure appears as Figure 2 in Appendix D.

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

## A  SPARSIFIER BACKGROUND AND RELATED WORK

Rafiey & Yoshida (2022) were the first to show how to construct a *sparsifier* for $F$. That is, given a parameter $\epsilon > 0$ they show how to find a vector $w \in \mathbb{R}^N$ such that the number of non-zero elements in $w$ is small in expectation and the function $\hat{F} = \sum_{i=1}^{N} w_i f^i$ satisfies with high probability (w.h.p)[9] that $\forall S \subseteq E, (1 - \epsilon)F(S) \leq \hat{F}(S) \leq (1 + \epsilon)F(S)$.

Specifically, every $f^i$ is sampled with probability $\alpha_i$ proportional to $p_i = \max_{S \subseteq E, F(S) \neq 0} \frac{f^i(S)}{F(S)}$. If it is sampled, it is included in the sparsifier with weight $1/\alpha_i$, which implies that $\mathbb{E}[w_i] = 1$. While calculating the $p_i$'s exactly requires exponential time, Rafiey & Yoshida (2022) make do with an approximation, which can be calculated using interior point methods (Bai et al., 2016).

Rafiey & Yoshida (2022) show that if all $f^i$'s are non-negative and monotone[10], the above sparsifier can be constructed by an algorithm that requires $poly(N)$ oracle evaluations and the sparsifier will have expected size $O(\epsilon^{-2}Bn^{2.5} \log n)$, where $B = \max_{i \in [N]} B_i$ and $B_i$ is the number of extreme points in the base polyhedron of $f^i$. They extend their results to matroid constraints of rank $r$ and show that a sparsifier with expected size $O(\epsilon^{-2}Brn^{1.5} \log n)$ can be constructed.

For the specific case of a cardinality constraint $k$, this implies a sparsifier of expected size $O(\epsilon^{-2}Bkn^{1.5} \log n)$ can be constructed using $poly(N)$ oracle evaluations. The sparsifier construction is treated as a *preprocessing step*, and therefore the actual execution of Greedy on the sparsifier requires only $O(\epsilon^{-2}Bk^2n^{2.5} \log n)$ oracle evaluations to get a $(1 - 1/e - \epsilon)$ approximation. This is an improvement over Greedy when $N \gg n, B$.

Recently, Kudla & Zivný (2023) showed improved results for the case of *bounded curvature*. The *curvature* of a submodular function $F$ is defined as $c = 1 - \min_{S \subseteq E, e \in E \setminus S} \frac{F_S(e)}{F(e)}$. We say that $F$ has *bounded-curvature* if $c < 1$. Submodular functions with bounded curvature (Conforti & Cornuéjols, 1984) offer a balance between modularity and submodularity, capturing the essence of diminishing returns without being too extreme.

They show that when the curvature of all $f^i$'s and of $F$ is constant, it is possible to reduce the preprocessing time to $O(Nn)$ oracle queries and to reduce the size of the sparsifier by a factor of $\sqrt{n}$. Furthermore, their results extend to the much more general case of $k$-*submodular functions*. While this significantly improves over the number of oracle calls compared to (Rafiey & Yoshida, 2022), the runtime of the preprocessing step depends on $\log \left( \max_{i \in [N]} \frac{\max_{e \in E} f^i(e)}{\min_{e \in E, f^i(e) > 0} f^i(e)} \right)$.

Note that the $B$ factor in (Kudla & Zivný, 2023; Rafiey & Yoshida, 2022) can be exponential in $n$ in the worst case.

The current state of the art is due to Kenneth & Krauthgamer (2023) where they show that by sampling according to $p_i = \max_{e \in E, F(e) \neq 0} \frac{f^i(e)}{F(e)}$ it is possible to get both a fast sparsifier construction time of $O(Nn)$ and a small sparsifier size of $O(\frac{n^3}{\epsilon^2})$. Their analysis also implies that if the solution size is bounded by $k$ (e.g., a cardinality constraint), a sparsifier of size $O(\frac{nk^2}{\epsilon^2})$ is sufficient. They also present results for general submodular functions, however, we only focus on their results for monotone functions, which are relevant for this paper.

## B  $p$-SYSTEMS

$p$-**systems**    The concept of $p$-systems offers a generalized framework for understanding independence families, parameterized by an integer $p$. We can define a $p$-system in the context of an independence family $\mathcal{I} \subseteq 2^E$ and $E' \subseteq E$. Let $\mathcal{B}(E')$ be the maximal independent sets within $\mathcal{I}$ that are also subsets of $E'$. Formally, $\mathcal{B}(E') = \{A \in \mathcal{I} | A \subseteq E' \text{ and no } A' \in \mathcal{I} \text{ exists s.t } A \subset A' \subseteq E'\}$.

---

[9]Probability at least $1 - 1/n^c$ for an arbitrary constant $c > 1$. The value of $c$ does not affect the asymptotics of the results we state (including our own).

[10]Rafiey & Yoshida (2022) also present results for non-monotone functions, however, Kudla & Zivný (2023) point out an error in their calculation and note that the results only hold when all $f^i$'s are monotone.

A distinguishing characteristic of a $p$-system is that for every $E' \subseteq E$, the ratio of the sizes of the largest to the smallest sets in $\mathcal{B}(E')$ does not exceed $p$: $\frac{\max_{A \in \mathcal{B}(E')} |A|}{\min_{A \in \mathcal{B}(E')} |A|} \leq p$.

The significance of $p$-systems lies in their ability to encapsulate a variety of combinatorial structures. For instance, when the intersection of $p$ matroids can be described using $p$-systems. In graph theory, the collection of matchings in a standard graph can be viewed as a 2-system. Extending this to hypergraphs, where edges might have cardinalities up to $p$, the set of matchings therein can be viewed as a $p$-system.

**The greedy algorithm for $p$-systems**     Formally, the optimization problem can be expressed as: $\max_{S \in \mathcal{I}} F(S)$ where the pair $(E, \mathcal{I})$ characterizes a $p$-system and $F : 2^E \to \mathbb{R}^+$ denotes a non-negative monotone submodular set function. It was shown by Nemhauser et al. (1978) that the natural greedy approach achieves an optimal approximation ratio of $\frac{1}{p+1}$. Setting $A_j = \{e \mid S_j + e \in \mathcal{I}\}$ (i.e., $S_j$ remains an independent set after adding $e$) in Algorithm 1 we get the greedy algorithm of Nemhauser et al. (1978). Note that for general $p$-systems it might be that $k = n$, however, there are very natural problems where $k \ll n$. For example, for maximum matching, $E$ corresponds to all edges in the graph, which can be quadratic in the number of nodes, while the solution is at most linear in the number of nodes.

## C   MISSING PROOFS

**Proof of Theorem 2.2**     Theorem 2.2 directly follows from the two lemmas below.

**Lemma C.1.** *Let $\epsilon' \leq \epsilon/2k$. Algorithm 1 with an additive $\epsilon'$-approximate incremental oracle achieves a $(1 - 1/e - \epsilon)$-approximation under a cardinality constraint $k$.*

*Proof.* Let $S^*$ be some optimal solution for $F$. We start by proving that the following holds for every $j \in [k]$:

$$F(S_{j+1}) - F(S_j) \geq \frac{1}{k}((1 - \epsilon)F(S^*) - F(S_j))$$

Fix some $j \in [k]$ and let $S^* \setminus S_j = \{e_1^*, \ldots, e_\ell^*\}$ where $\ell \leq k$. Let $S_t^* = \{e_1^*, \ldots, e_t^*\}$, and $S_0^* = \emptyset$. Let us first use submodularity and monotonicity to upper bound $F(S^*)$.

$$F(S^*) \leq F(S^* + S_j)$$
$$= F(S_j) + \sum_{t=1}^{\ell}[F(S_j + S_t^*) - F(S_j + S_{t-1}^*)]$$
$$\leq F(S_j) + \sum_{t=1}^{\ell} F_{S_j}(e_t^*) \leq F(S_j) + \sum_{t=1}^{\ell} \max_{e \in E \setminus S_j} F_{S_j}(e)$$
$$\leq F(S_j) + k \max_{e \in E \setminus S_j} F_{S_j}(e)$$
$$\leq F(S_j) + k(\max_{e \in E \setminus S_j} \hat{F}_{S_j}^j(e) + \epsilon' F(S^*))$$

Where the last inequality is due to the fact that $\hat{F}_{S_j}^j$ is an additive $\epsilon'$-approximate incremental oracle.

Noting that $e_j = \arg\max_{e \in E \setminus S_j} \hat{F}_{S_j}^j(e)$ we get that:

$$F(S^*) \leq F(S_j) + k(\hat{F}_{S_j}^j(e_j) + \epsilon' F(S^*))$$
$$\implies \hat{F}_{S_j}^j(e_j) \geq \frac{1}{k}((1 - \epsilon' k)F(S^*) - F(S_j))$$

The above lower bounds the progress on the $j$-th mini-batch. Now, let us bound the progress on $F$. Again, we use the fact that $\hat{F}^j_{S_j}$ is an additive $\epsilon'$-approximate incremental oracle.

$$F(S_{j+1}) - F(S_j) \geq \hat{F}^j_{S_j}(e_j) - \epsilon' F(S^*)$$

$$\geq \frac{1}{k}((1 - \epsilon'k)F(S^*) - F(S_j)) - \epsilon' F(S^*)$$

$$\geq \frac{1}{k}((1 - 2\epsilon'k)F(S^*) - F(S_j))$$

Finally, using the fact that $\epsilon' \leq \epsilon/2k$ we get:

$$F(S_{j+1}) - F(S_j) \geq \frac{1}{k}((1 - \epsilon)F(S^*) - F(S_j))$$

Rearranging, the result directly follows using standard arguments.

$$F(S_{k+1}) > \frac{(1 - \epsilon)}{k} F(S^*) + (1 - \frac{1}{k})F(S_k)$$

$$\geq \frac{(1 - \epsilon)}{k} F(S^*)(\sum_{i=0}^{k}(1 - \frac{1}{k})^i) + F(\emptyset)$$

$$\geq F(S^*) \frac{(1 - \epsilon)(1 - \frac{1}{k})^k}{k(1 - (1 - \frac{1}{k}))} = (1 - \epsilon)(1 - \frac{1}{k})^k F(S^*)$$

$$\geq (1 - \epsilon)(1 - 1/e)F(S^*) \geq (1 - 1/e - \epsilon)F(S^*)$$

$\square$

**Lemma C.2.** *Let $\epsilon' \leq \epsilon/2kp$. Algorithm 1 with an additive $\epsilon'$-approximate incremental oracle achieves a $(\frac{1-\epsilon}{1+p})$-approximation under a $p$-system constraint.*

*Proof.* Let $S^*$ be some optimal solution for $F$.

Assume without loss of generality that the solution returned by the algorithm consists of $k$ elements $S_{k+1} = \{e_1, \ldots, e_k\}$.

We show the existence of a partition $S^*_1, S^*_2, \ldots, S^*_k$ of $S^*$ such that $F_{S_j}(e_j) \geq \frac{1}{p}F_{S_{k+1}}(S^*_j) - 2\epsilon' F(S^*)$. Note, we allow some of the sets in the partition to be empty.

Define $T_k = S^*$. For $j = k, k-1, ..., 2$ execute: Let $B_j = \{e \in T_j \mid S_j + e \in \mathcal{I}\}$. If $|B_j| \leq p$ set $S^*_j = B_j$; else pick an arbitrary $S^*_j \subset B_j$ with $|S^*_j| = p$. Then set $T_{j-1} = T_j \setminus S^*_j$ before decreasing $j$. After the loop set $S^*_1 = T_1$. It is clear that for $j = 2, ..., k, |S^*_j| \leq p$.

We prove by induction over $j = 0, 1, ..., k-1$ that $|T_{k-j}| \leq (k - j)p$. For $j = 0$, when the greedy algorithm stops, $S_{k+1}$ is a maximal independent set contained in $E$, therefore any independent set (including $T_k = S^*$) satisfies $|T_k| \leq p|S_{k+1}| = pk$. We proceed to the inductive step for $j > 0$. There are two cases: (1) $|B_{k-j+1}| > p$, which implies that $\left|S^*_{k-j+1}\right| = p$ and using the induction hypothesis we get that $|T_{k-j}| = |T_{k-j+1}| - \left|S^*_{k-j+1}\right| \leq (k - j + 1)p - p = (k - j)p$. (2) $|B_{k-j+1}| \leq p$, it holds that $T_{k-j} = T_{k-j+1} \setminus B_{k-j+1}$. Let $Y = S_{k-j+1} + T_{k-j}$. Due to the definition of $B_{k-j+1}$ it holds that $S_{k-j+1}$ is a maximal independent set in $Y$. It holds that $T_{k-j}$ is independent and contained in $Y$, therefore $|T_{k-j}| \leq p|S_{k-j+1}| = p(k - j)$.

Finally, we get that $|T_1| = |S^*_1| \leq p$. By construction it holds that $\forall j \in [k], \forall e \in S^*_j, S_j + e$ is independent. From the choice made by the greedy algorithm and the fact that $\hat{F}^j_{S_j}$ is an additive $\epsilon'$-approximate incremental oracle, it follows that for each $e \in S^*_j$:

$$F_{S_j}(e_j) \geq \hat{F}^j_{S_j}(e_j) - \epsilon' F(S^*)$$

$$\geq \hat{F}^j_{S_j}(e) - \epsilon' F(S^*) \geq F_{S_j}(e) - 2\epsilon' F(S^*)$$

Hence,

$$\left|S_j^*\right| F_{S_j}(e_j) \geq \sum_{e \in S_j^*} \left(F_{S_j}(e) - 2\epsilon' F(S^*)\right)$$

$$\geq F_{S_j}(S_j^*) - 2\epsilon' \left|S_j^*\right| F(S^*) \geq F_{S_{k+1}}(S_j^*) - 2\epsilon' \left|S_j^*\right| F(S^*)$$

Using submodularity in the last two inequalities.

For all $j \in \{1, 2, ..., k\}$ it holds that $\left|S_j^*\right| \leq p$, and thus $F_{S_j}(e_j) \geq \frac{1}{p} F_{S_{k+1}}(S_j^*) - 2\epsilon' F(S^*)$. Using the partition, we get that:

$$F(S_{k+1}) \geq \sum_{j=1}^{k} F_{S_j}(e_j) \geq \sum_{j=1}^{k} \left(\frac{1}{p} F_{S_{k+1}}(S_j^*) - 2\epsilon' F(S^*)\right)$$

$$\geq \frac{1}{p} F_{S_{k+1}}(S^*) - 2\epsilon' k F(S^*)$$

$$\geq \frac{1}{p} (F(S^*) - F(S_{k+1})) - 2\epsilon' k F(S^*)$$

Where the second to last inequality is due to submodularity, and the last is due to monotonicity. Rearranging, we get that:

$$F(S_{k+1}) \geq \frac{(1 - 2p\epsilon'k)}{p+1} F(S^*)$$

As $\epsilon' < \frac{\epsilon}{2pk}$ we get the desired result. $\qquad\square$

## D ADDITIONAL EXPERIMENTS

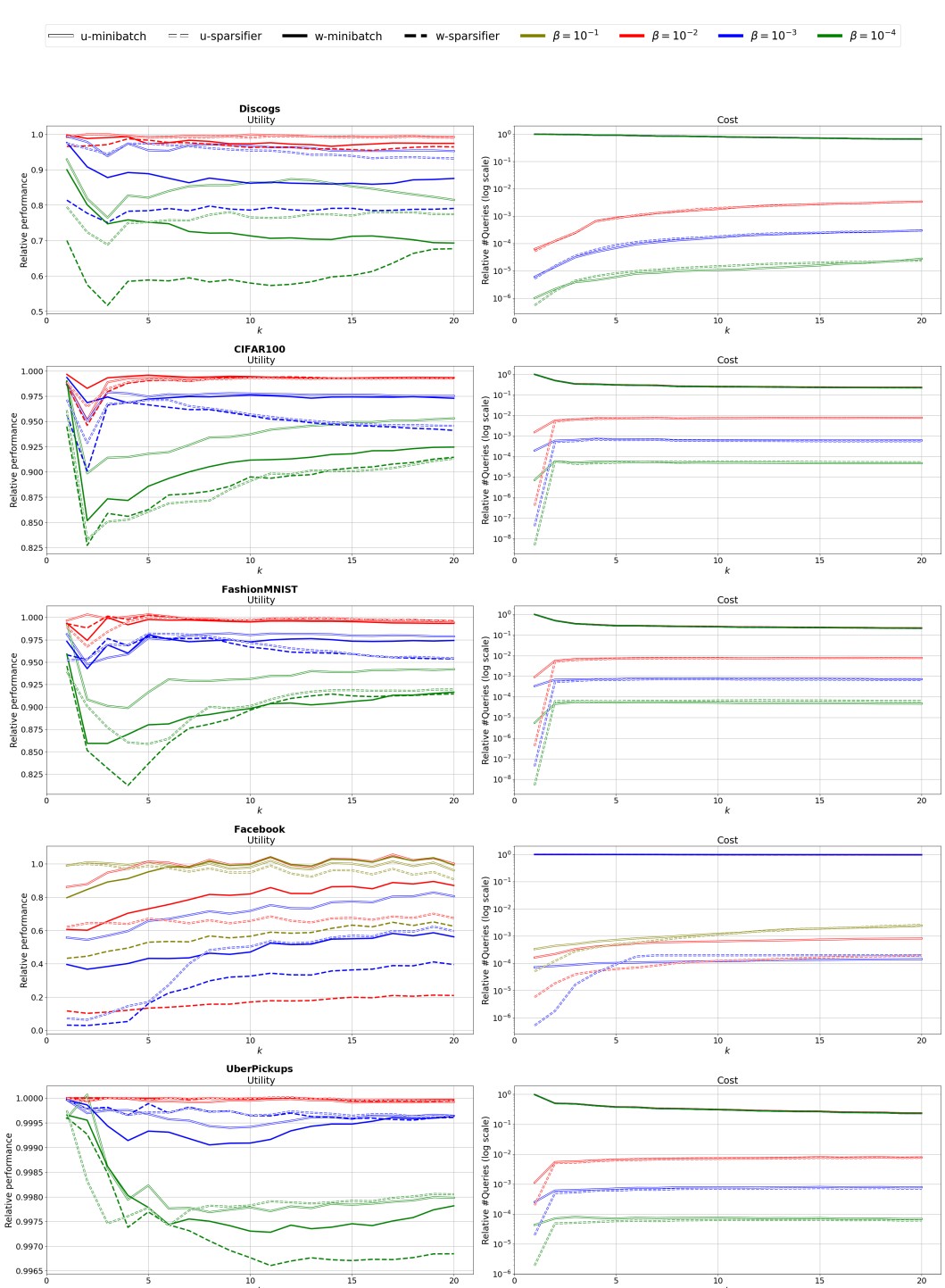

Figure 2: A larger version of Figure 1. Sparsifier and mini-batch compared using lazy-greedy. We present plots for $\beta \in \left\{10^{-2}, 10^{-3}, 10^{-4}\right\}$ for all datasets except Facebook where we plot $\beta \in \left\{10^{-1}, 10^{-2}, 10^{-3}\right\}$ due to its small size.

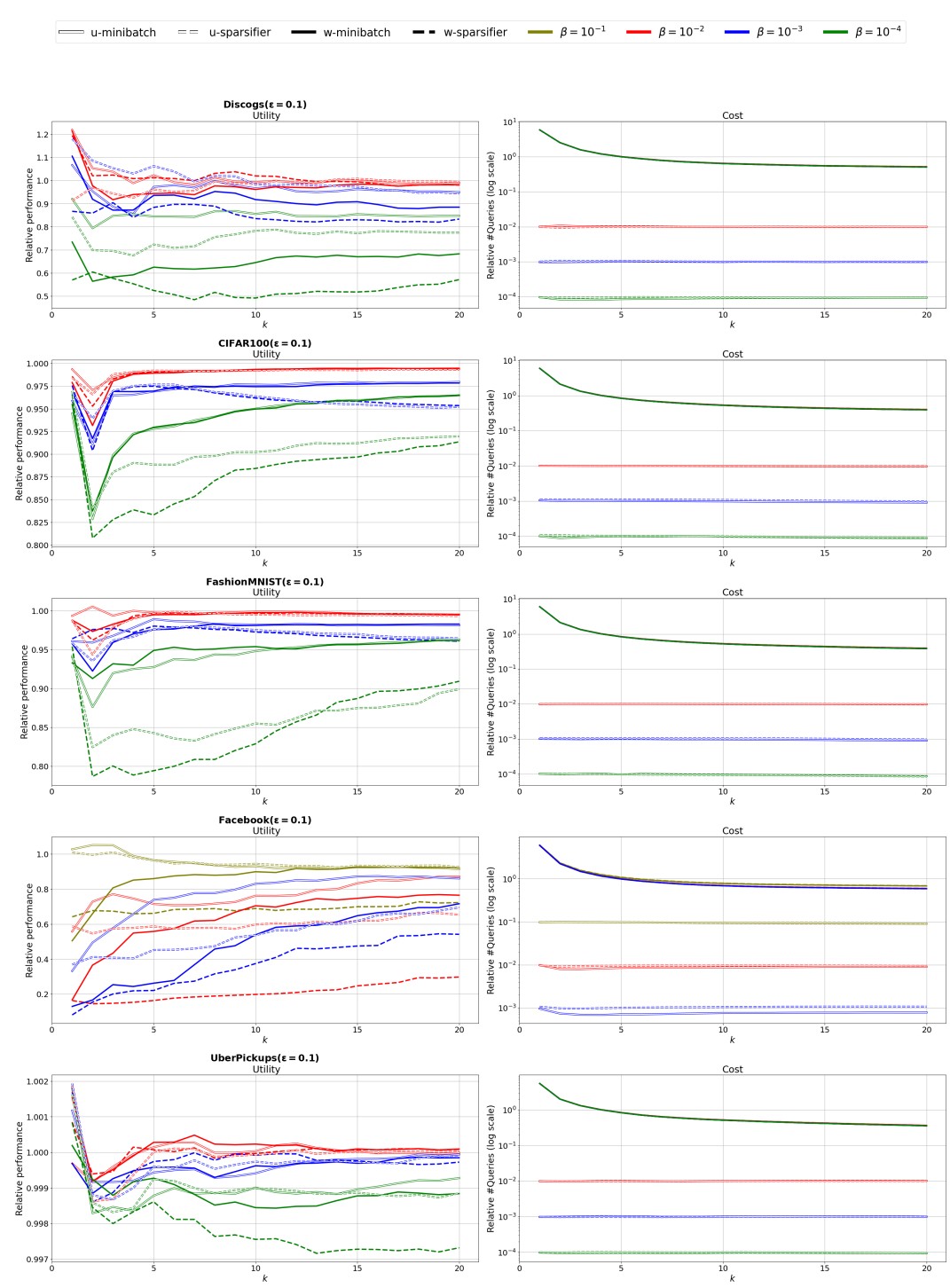

Figure 3: Sparsifier and mini-batch compared with stochastic-greedy for $\epsilon = 0.1$. We present plots for $\beta \in \left\{ 10^{-2}, 10^{-3}, 10^{-4} \right\}$ for all datasets except Facebook where we plot $\beta \in \left\{ 10^{-1}, 10^{-2}, 10^{-3} \right\}$ due to its small size.

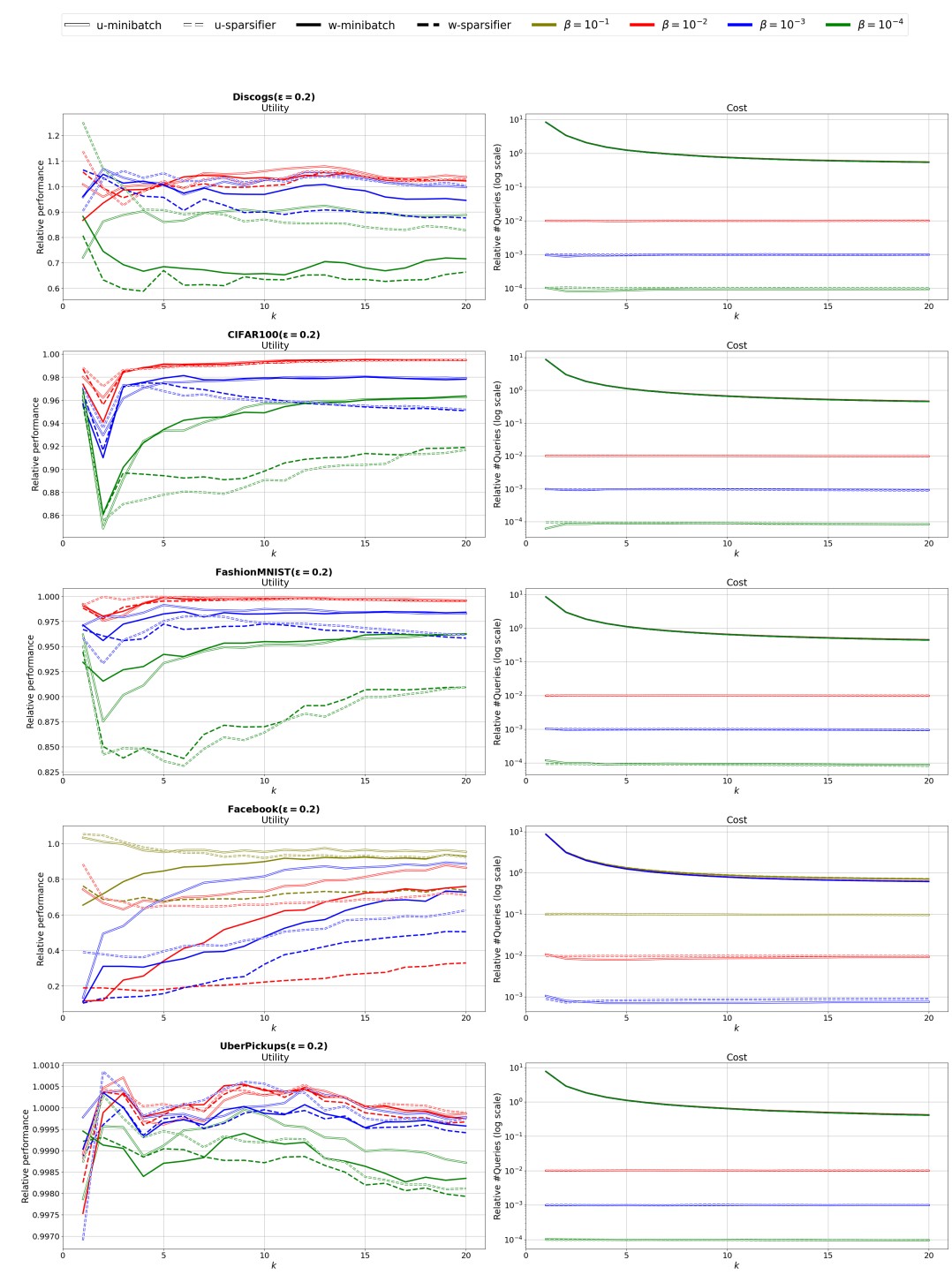

Figure 4: Sparsifier and mini-batch compared with stochastic-greedy for $\epsilon = 0.2$. We present plots for $\beta \in \left\{ 10^{-2}, 10^{-3}, 10^{-4} \right\}$ for all datasets except Facebook where we plot $\beta \in \left\{ 10^{-1}, 10^{-2}, 10^{-3} \right\}$ due to its small size.

# E  THEORETICAL GUARANTEES FOR THE UNIFORM SPARSIFIER

Let us start with a definition.

**Definition E.1.** We say that $\hat{F}$ is an $\epsilon$-*additive sparsifier* for $F$ if $\forall S \subseteq E, \max\{F(S) - \epsilon F(S^*), 0\} \leq \hat{F}(S) \leq F(S) + \epsilon F(S^*)$, where $S^*$ is an optimal solution for $F$.

We state the following theorem (note that it does not rely on our smoothing assumptions).

**Theorem E.2.** *Let $\hat{F}$ be an $\epsilon$-additive sparsifier for $F$, and $\hat{S}$ be a $\psi$-approximate solution for $\hat{F}$, for $\psi \in (0, 1]$. Then $\hat{S}$ is a $(\psi - 2\epsilon)$-approximate solution for $F$.*

*Proof.* Let $\hat{S}^*, S^*$ be optimal solutions for $\hat{F}, F$ respectively, and let $\hat{S}$ be a $\psi$-approximate solution for $\hat{F}$. It holds that

$$F(\hat{S}) \geq \hat{F}(\hat{S}) - \epsilon F(S^*) \geq \psi \hat{F}(\hat{S}^*) - \epsilon F(S^*) \geq \psi \hat{F}(S^*) - \epsilon F(S^*)$$
$$\geq \psi(F(S^*) - \epsilon F(S^*)) - \epsilon F(S^*) = (\psi - \psi\epsilon - \epsilon)F(S^*) \geq (\psi - 2\epsilon)F(S^*)$$

Where the first and fourth transitions use the definition of the sparsifier, the second uses the fact that $\hat{S}$ is an approximate solution for $\hat{F}$ and the third transition uses the fact that $\hat{S}$ is an optimal solution for $\hat{F}$. $\qquad\square$

An analogue proof also works for the case where we aim to minimize $F$. Furthermore, the above holds independently of any constraints.

Next we show that under the smoothing assumptions we can construct an $\epsilon$-additive sparsifier using uniform sampling. Specifically, for a parameter $\alpha \leq N$ we set $w_i = N/\alpha$ with probability $\alpha/N$ and 0 otherwise. The sparsifier is $\hat{F} = \sum_{i=1}^{N} w_i f^i$. We prove an analogue to Lemma 3.4.

**Lemma E.3.** *Under the smoothing assumptions, for every $S \subseteq E$ ($\hat{F}$ sampled after $S$ is fixed) and $\mu \geq F(e)$, it holds that $\mathbb{P}\left[|\hat{F}(S) - F(S)| \geq \epsilon\mu\right] \leq 2\exp\left(-\frac{\epsilon^2\alpha\mu}{6F(e^*)}\right)$.*

*Proof.* Fix some $S \subseteq E$. Due to the fact that $\forall i \in [N], \mathbb{E}[w_i] = 1$ we have $\mathbb{E}[\hat{F}(S)] = \mathbb{E}[\sum_{i \in [N]} w_i f^i(S)] = F(S)$. Applying a Chernoff bound (Theorem 3.1) we have

$$\mathbb{P}\left[|\hat{F}(S) - F(S)| \geq \epsilon\mu\right] \leq 2\left(-\frac{\epsilon^2\mu}{3a}\right)$$

where $a = \max\{\frac{N}{\alpha}f^i(S)\}_{i \in [N]} \leq N/\alpha$.

Let us upper bound $a$ as a function of $F(e^*)$ where $e^* \in E$ is the element guaranteed by the smoothing assumption. Rearranging Lemma 3.3 we get that $N \leq 2F(e^*)/\phi$. We conclude that $a \leq N/\alpha \leq 2F(e^*)/\phi\alpha$. Given the above upper bound for $a$ we get:

$$\mathbb{P}\left[|\hat{F}(S) - F(S)| \geq \epsilon\mu\right] \leq 2\left(-\frac{\epsilon^2\mu}{3a}\right) \leq 2\exp\left(-\frac{\epsilon^2\alpha\mu}{6F(e^*)}\right)$$

$\qquad\square$

We are now ready to prove the main theorem for this section.

**Theorem E.4.** *Under the smoothing assumptions, for any $\gamma > 0$ and $\alpha = \Theta(k\gamma\epsilon^{-2}\log n)$, it holds that $\hat{F}$ is an $(\epsilon F(e^*)/\gamma F(S^*))$-additive sparsifier for $F$ of size $\Theta(k\gamma\epsilon^{-2}\log n)$ in expectation and w.h.p, where $k \leq n$ is an upper bound on any solution for $F$.*

*Proof.* Let us prove the approximation guarantees. We break the analysis into cases.

$F(e^*) \leq \gamma F(S)$**:** Setting $\mu = F(S)$ we get:

$$\mathbb{P}\left[|\hat{F}(S) - F(S)| \geq \epsilon F(S)\right] \leq 2\exp\left(-\frac{\epsilon^2\alpha F(S)}{6F(e^*)}\right) \leq 2\exp\left(-\frac{\epsilon^2\alpha}{6\gamma}\right) \leq 1/n^{k+3}$$

$F(e^*) > \gamma F(S)$:     Here we can set $\mu = F(e^*)/ \geq F(S)$ and get:

$$\mathbb{P}\left[|\hat{F}(S) - F(S)| \geq \epsilon F(e^*)/\gamma\right] \leq 2\exp\left(-\frac{\epsilon^2 \alpha F(e^*)}{6\gamma F(e^*)}\right) \leq 2\exp\left(-\frac{\epsilon^2 \alpha}{6\gamma}\right) \leq 1/n^{k+3}$$

Where in both cases, the last inequality is by setting an appropriate constant in $\alpha = \Theta(k\gamma\epsilon^{-2}\log n)$. We take a union bound over all $S \subseteq E$ such that $|S| \leq k$ (at most $2^k$) and $j \in [k]$ (at most $n^2$ values), which concludes the proof.

$\square$

While the above yields improved approximation guarantees for arbitrary constraints and arbitrary algorithms, in this paper we focus on the greedy algorithm under cardinality and $p$-system constraints. Let us state our results for these two constraints by combining Theorem E.2 and Theorem E.4. The result for a $p$-system constraint follows via a similar case-based analysis as Theorem 2.4.

**Theorem E.5.** *Under the smoothing assumptions, when applied to maximize a non-negative monotone submodular function, the greedy algorithm executed on $\hat{F}$ constructed with parameter $\alpha = \Theta(k\epsilon^{-2}\log n)$ achieves w.h.p a $(1 - 1/e - \epsilon)$-approximation under a cardinality constraint and a $(\frac{1-\epsilon}{1+p})$-approximation under a $p$-system constraint using $O(\frac{k^2 n \log n}{\epsilon^2 \phi})$ oracle evaluations in expectation and w.h.p.*

# F    WEIGHTED MINI-BATCH GREEDY ALGORITHM

In this section, we define the weighted mini-batch algorithm and prove *worst-case* guarantees, which are superior to those of the sparsifier-based approach.

The weighted mini-batch algorithm is identical to the uniform version, except that we sample according to the following probabilities $\forall i \in [N], p_i = \max_{e \in E, F(e)\neq 0} \frac{f^i(e)}{F(e)}$, which are the same as in the sparsifier algorithm of (Kenneth & Krauthgamer, 2023). That is, instead of sampling a single $\hat{F}$ at the beginning, we sample a new $\hat{F}^j$ (*mini-batch*), for every step of the algorithm.

Similar to the uniform version, we show that when $\hat{F}^j_{S_j}$ is sampled using mini-batch sampling, we indeed get, w.h.p, an (additive) approximate incremental oracle for every step of the algorithm. We present our sampling procedure in Algorithm 3 and the complete mini-batch algorithm in Algorithm 4.

---

**Algorithm 3: Sample($\alpha, \{p_i\}_{i=1}^N$)**

1   $w \leftarrow \vec{0}$
2   **for** $i = 1$ *to* $N$ **do**
3      $\alpha_i \leftarrow \min\{1, \alpha p_i\}$
4      $w_i \leftarrow 1/\alpha_i$ with probability $\alpha_i$
5   **end**
6   return $\hat{F} = \sum_{i=1}^N w_i f^i$

---

**Algorithm 4: Weighted mini-batch greedy**

1   $\forall i \in [N], p_i \leftarrow \max_{e \in E, F(e)\neq 0} \frac{f^i(e)}{F(e)}$
2   $\alpha$ is a batch parameter
3   $S_1 \leftarrow \emptyset$
4   $k$ is an upper bound on the size of the solution
5   **for** $j = 1$ *to* $k$ **do**
6      Let $A_j \subseteq E \setminus S_j$
7      **if** $A_j = \emptyset$ **then** return $S_j$
8      Let $\hat{F}^j \leftarrow Sample(\alpha, \{p_i\}_{i=1}^N)$
9      $e_j = \arg\max_{e \in A_j} \hat{F}^j_{S_j}(e)$
10     $S_{j+1} = S_j + e_j$
11   **end**
12   return $S_{k+1}$

---

We analyze the relation between the batch parameter, $\alpha$, and the type of approximate incremental oracles guaranteed by our sampling procedure.

We start by with the following lemma from (Kenneth & Krauthgamer, 2023), which bounds the expected size of $\hat{F}$. We present a proof for completeness.

**Lemma F.1.** *The expected size of $\hat{F}$ is $\alpha \sum_{i=1}^N p_i \leq \alpha n$.*

*Proof.* We start by showing that $\sum_{i=1}^{N} p_i \leq n$. Let us divide the range $[N]$ into

$$A_e = \left\{ i \in N \mid e = \underset{e' \in E, F(e') \neq 0}{\arg \max} \frac{f^i(e')}{F(e')} \right\}$$

If 2 elements in $E$ achieve the maximum value for some $i$, we assign it to a single $A_e$ arbitrarily.

$$\sum_{i=1}^{N} p_i = \sum_{i=1}^{N} \max_{e \in E} \frac{f^i(e)}{F(e)} = \sum_{e \in E} \sum_{i \in A_e} \frac{f^i(e)}{F(e)} = \sum_{e \in E} \frac{\sum_{i \in A_e} f^i(e)}{F(e)} \leq \sum_{e \in E} 1 \leq n$$

Let $X_i$ be an indicator variable for the event $w_i > 0$. We are interested in $\sum_{i=1}^{N} \mathbb{E}[X_i]$. It holds that:

$$\sum_{i=1}^{N} \mathbb{E}[X_i] = \sum_{i=1}^{N} \alpha_i \leq \sum_{i=1}^{N} \alpha p_i = \alpha \sum_{i=1}^{N} p_i \leq \alpha n$$

$\square$

The following lemma provides concentration guarantees for $\hat{F}$ in Algorithm 3.

**Lemma F.2.** *For every $S \subseteq E$ ($\hat{F}$ sampled after $S$ is fixed) and for every $e \in E$ and $\mu \geq F_S(e)$, it holds that* $\mathbb{P}\left[ |\hat{F}_S(e) - F_S(e)| \geq \epsilon \mu \right] \leq 2 \exp\left( -\frac{\epsilon^2 \alpha \mu}{3F(e)} \right)$.

*Proof.* Fix some $e \in E$. Let $G = \sum_{i \in I} f^i$, where $I = \{ i \in [N] \mid \alpha_i = 1 \}$. Let $F'_S(e) = F_S(e) - G_S(e)$ and $\hat{F}'_S(e) = \hat{F}_S(e) - G_S(e)$. Let $J = [N] \setminus I$. It holds that:

$$\mathbb{P}\left[ |\hat{F}_S(e) - F_S(e)| \geq \epsilon \mu \right] = \mathbb{P}\left[ |\hat{F}'_S(e) + G_S(e) - F'_S(e) - G_S(e)| \geq \epsilon \mu \right] = \mathbb{P}\left[ |\hat{F}'_S(e) - F'_S(e)| \geq \epsilon \mu \right]$$

Due to the fact that $\mathbb{E}[w_i] = 1$ we have $\mathbb{E}[\hat{F}'_S(e)] = \mathbb{E}[\sum_{i \in J} w_i f^i_S(e)] = F'_S(e)$. As $f^i$'s are monotone, it holds that $\mu \geq F_S(e) \geq F'_S(e)$. Applying a Chernoff bound (Theorem 3.1) we have

$$\mathbb{P}\left[ |\hat{F}'_S(e) - F'_S(e)| \geq \epsilon \mu \right] \leq 2 \exp\left( -\epsilon^2 \mu / 3a \right)$$

where $a = \max\{w_i f^i_S(e)\}_{i \in J}$. Recall that $w_i = 1/\alpha_i$ where $\alpha_i = \min\{1, \alpha p_i\}$ and $\alpha_i < 1$ for all $i \in J$. Let us upper bound $a$.

$$a = \max_{i \in J} w_i f^i_S(e) = \max_{i \in J} \frac{f^i_S(e)}{\alpha p_i} = \max_{i \in J} \frac{f^i_S(e)}{\alpha \cdot \max_{e' \in E} \frac{f^i(e')}{F(e')}} \leq \max_{i \in J} \frac{f^i(e)}{\alpha \cdot \frac{f^i(e)}{F(e)}} = \frac{F(e)}{\alpha}$$

Where the inequality is due to submodularity and non-negativity in the numerator and maximality in the denominator. Note that the above is also correct if we only have some approximation to $p_i$ – i.e., given $p'_i > p_i \lambda$ for $\lambda \in (0, 1)$, we can increase $\alpha$ by a $1/\lambda$ factor and the above still holds. Given the above upper bound for $a$ we get:

$$\mathbb{P}\left[ |\hat{F}_S(e) - F_S(e)| \geq \epsilon \mu \right] \leq 2 \left( -\frac{\epsilon^2 \mu}{3a} \right) \leq 2 \exp\left( -\frac{\epsilon^2 \alpha \mu}{3F(e)} \right)$$

$\square$

We state the main theorem for the section below.

**Theorem F.3.** *The weighted mini-batch greedy algorithm (Algorithm 4) maximizing a non-negative monotone submodular function has the following guarantees:*

1. *If $F$ has curvature bounded by $c$, and $\alpha = \Theta(\frac{\log n}{\epsilon^2 (1-c)})$ it holds w.h.p that $\forall j \in [k]$ that $\hat{F}^j_{S_j}$ is an $(1 - \epsilon)$-approximate incremental oracle.*

2. *If $\alpha = \Theta(\epsilon^{-2} \gamma \log n)$ it holds w.h.p that $\forall j \in [k]$ that $\hat{F}^j_{S_j}$ is an additive $(\epsilon/\gamma)$-approximate incremental oracle, for any parameter $\gamma > 0$.*

*Furthermore, the number of oracle evaluations during preprocessing is $O(nN)$ and an expected $\alpha(\sum_{i=1}^{N} p_i)(\sum_{j=1}^{k} |A_j|) = O(\alpha k n^2)$ during execution.*

Combining Theorem F.3 (setting $\gamma = k$ for a cardinality constraint and $\gamma = kp$ for a $p$-system constraint) with Theorem 2.1 and Theorem 2.2 we state our main result.

**Theorem F.4.** *The weighted mini-batch greedy algorithm maximizing a non-negative monotone submodular function requires $O(nN)$ oracle calls during preprocessing and has the following guarantees:*

1. *If F has curvature bounded by c, it achieves w.h.p a $(1 - 1/e - \epsilon)$-approximation under a cardinality constraint and $(\frac{1-\epsilon}{1+p})$-approximation under a $p$-system constraint with an expected $O(\frac{kn^2 \log n}{\epsilon^2(1-c)})$ oracle evaluations for both cases.*

2. *It achieves w.h.p a $(1 - 1/e - \epsilon)$-approximation under a cardinality constraint and $(\frac{1-\epsilon}{1+p})$-approximation under a $p$-system constraint with an expected $O(k^2(n/\epsilon)^2 \log n)$ and $O(k^2 p(n/\epsilon)^2 \log n)$ oracle evaluations respectively.*

*Proof.* The number of oracle evaluations is due to Lemma F.1 and the fact that the algorithm executes for $k$ iteration and must evaluate $|A_j| \leq n$ elements per iteration.

Let us prove the approximation guarantees. Let us start with the bounded curvature case. Fix some $S_j$. As $\hat{F}^j$ is sampled after $S_j$ is fixed, we can fix some $e \in E$ and apply Lemma 3.4 with $\mu = F_{S_j}(e)$. We get that:

$$\mathbb{P}\left[|\hat{F}_{S_j}^j(e) - F_{S_j}(e)| \geq \epsilon F_{S_j}(e)\right] \leq 2 \exp\left(-\frac{\epsilon^2 \alpha F_{S_j}(e)}{3F(e)}\right) \leq 2 \exp\left(-\frac{\epsilon^2 \alpha (1-c)}{3}\right) \leq 1/n^3$$

Where the second inequality is because $F_{S_j}(e)/F(e) \geq \min_{S \subseteq E, e' \in E \setminus S} F_S(e')/F(e') = 1 - c$, and the last transition is by setting an appropriate constant in $\alpha = \Theta(\frac{\log(n)}{\epsilon^2(1-c)})$. When the curvature is not bounded, we break the analysis into cases.

$F(e) \leq \gamma F_{S_j}(e)$**:**   Setting $\mu = F_{S_j}(e)$ we get:

$$\mathbb{P}\left[|\hat{F}_{S_j}^j(e) - F_{S_j}(e)| \geq \epsilon F_{S_j}(e)\right] \leq 2 \exp\left(-\frac{\epsilon^2 \alpha F_{S_j}(e)}{3F(e)}\right) \leq 2 \exp\left(-\frac{\epsilon^2 \alpha}{3\gamma}\right) \leq 1/n^3$$

$F(e) > \gamma F_{S_j}(e)$**:**   Here we can set $\mu = F(e)/\gamma \geq F_{S_j}(e)$ and get:

$$\mathbb{P}\left[|\hat{F}_{S_j}^j(e) - F_{S_j}(e)| \geq \epsilon F(e)/\gamma\right] \leq 2 \exp\left(-\frac{\epsilon^2 \alpha F(e)}{3\gamma F(e)}\right) \leq 2 \exp\left(-\frac{\epsilon^2 \alpha}{3\gamma}\right) \leq 1/n^3$$

Where in both cases, the last inequality is by setting an appropriate constant in $\alpha = \Theta(\epsilon^{-2} \gamma \log n)$. Note that in the first case we get a $(1 - \epsilon)$-approximate incremental oracle and in the second an additive $(\epsilon/\gamma)$-approximate incremental oracle. The second case is the worst of the two for our analysis. For both bounded and unbounded curvature, we take a union bound over all $e \in E$ and $j \in [k]$ (at most $n^2$ values), which concludes the proof. □

