# OpenReview forum: "Mini-batch Submodular Maximization"
_ICLR.cc/2026/Conference — Submitted to ICLR 2026_

### Official Review · Reviewer_nDVU · 2025-10-29

**Soundness:** 3
**Presentation:** 2
**Contribution:** 1
**Rating:** 2
**Confidence:** 4

**Summary:**

This paper investigates the problem of submodular maximization with a monotone decomposable objective function, i.e., $f(S)=\sum_{i=1}^Nf^i(S)$. The authors propose an algorithm based on uniform mini-batch sampling, which empirically outperforms existing sparsifier-based methods. To explain this advantage, they conduct a smoothed analysis demonstrating that uniform sampling is theoretically superior to weighted sampling in both the mini-batch and sparsifier settings.

**Strengths:**

1. The paper addresses a well-motivated problem with broad practical relevance and numerous applications.
2. The theoretical analysis of the proposed submodular maximization algorithm is rigorous yet presented in a clear and accessible manner.

**Weaknesses:**

1. A major concern lies in the significance of the proposed approach. The large-scale optimization of decomposable submodular functions can be viewed as a special case of stochastic submodular maximization, or more generally, submodular maximization with i.i.d. bandit feedback, where uniform sampling corresponds to drawing samples from the stochastic distribution. From this perspective, the authors should compare their results against existing works in this broader setting (e.g., [1–3]). Moreover, the use of uniform sampling to approximate the objective function is typically considered a baseline method in the literature, which limits the novelty of this contribution. As a result, the overall advancement provided by the paper appears marginal.

2. The presentation of the smoothed analysis is unclear, making it difficult to follow. For instance, in the description of the smoothing model, the authors state that $f^i(e*)$  is a random variable, but the source of randomness is not explicitly explained. It remains ambiguous whether the randomness arises from the function $f^i$ itself or from the sampled element $e^*$?

[1] Singla, Adish, Sebastian Tschiatschek, and Andreas Krause. "Noisy submodular maximization via adaptive sampling with applications to crowdsourced image collection summarization."

[2] Wenjing Chen, Shuo Xing, and Victoria G Crawford. A threshold greedy algorithm for noisy submodular maximization. arXiv preprint arXiv:2312.00155, 2023.

[3] Karimi, M., Lucic, M., Hassani, H., & Krause, A. Stochastic submodular maximization: The case of coverage functions. Advances in Neural Information Processing Systems, 30.

**Questions:**

1. Could you provide a comparison between your algorithm and the one proposed in [1]? It appears that your method may be a straightforward adaptation of the algorithm in [1], obtained by replacing the TopK selection step with taking the same number of samples of the marginal gain for each element in the universe.
2. Could you elaborate on the specific technical contributions of this work? The problem of submodular maximization under noise has been extensively studied in prior literature, where analyses typically focus on algorithms with additive or multiplicative approximation errors and employ concentration inequalities to estimate the objective value. It would be helpful if you could clarify how your analysis or techniques differ from these existing approaches.

[1] Singla, Adish, Sebastian Tschiatschek, and Andreas Krause. "Noisy submodular maximization via adaptive sampling with applications to crowdsourced image collection summarization."

**Details Of Ethics Concerns:**

No.

---

> ### Author Response · Authors · 2025-11-15
>
> Thank you for your comments and questions.
>
> Weaknesses
>
> (1) Thanks for raising this point. This is indeed an interesting comparison to make. Let us use the running example from [1] going forward. We are given a massive set of images and we are looking for a set of $k$ images that best represent the massive set. This can be represented as a decomposable submodular function $F = \sum f^i$, where $f^i$ measures the similarity of an input image to the $i$-th image in the set. Our approach will optimize this problem by sampling mini-batches, while in [1] they assume they do not have direct access to $F$ but instead have access to some noisy proxy function (e.g., user feedback for a set of selected images).
>
> If we understood correctly, you are asking why not just consider the batch as the noisy proxy function for $F$, in which case the results of [1] should still go through. This can be done as indeed the results of [1] assume a more general setting. The downside is that they provide weaker results / require more assumptions. We detail this below:
>
> - The results of [1] only provide additive approximation guarantees while we achieve multiplicative approximation guarantees.
>
> - The query complexity of [1] depends on a gap parameter for $F$, while our results do not.
>
> - The results of [1] do not hold for $p$-systems.
>
> - To plug the mini-batch approach as a noisy estimator into [1] we still need to show concentration guarantees - this cannot be done without assuming our smoothing model.
>
> - While performing an exact query comparison might be a bit involved, we believe that plugging the mini-batch approach into [1] will result in a worse query complexity as every evaluation of the estimator function requires sampling a fresh mini-batch ([1] assumes results from the estimator are IID).
>
> (2) Let us contrast the smoothing model with the worst-case model.
>
> Worst case model: an adversary can choose all $f^i$ in an adversarial way. There is no constraint on how the functions are chosen.
>
> Our model: an adversary can choose a *distribution* from which every $f^i$ is sampled under some constraints. Note, if we do not add any constraints here then this is just equivalent to the worst case model. The adversary can choose a distribution where all of the density is concentrated on a single element. Our constraints force the adversary to choose non-pathological distributions. Formally, the adversary can choose any distribution for $f^i$ as long as the distributions have the two properties mentioned in our smoothing model.
>
> Crucially, each sampled $f^i$ is itself a deterministic function; the randomness comes from which deterministic function is drawn from the adversary-chosen distribution. Thus $f^i$ is a random variable with respect to this sampling process. The underlying probability space is therefore the product space generated by these adversarially chosen, but constrained, distributions.
>
>
> Models in the literature: Let us contrast our model with a commonly used model in the literature to highlight the generality of our model. We consider the widely studied Local Max-Cut problem (https://www.roeglin.org/publications/SODA14.pdf). Here the input is an edge weighted graph to which we want to apply smoothing. The standard smoothing model assumes that every edge weight is sampled from some distribution where the only constraint is that its density is upper bounded by $\phi$. Intuitively, this means that the adversary can choose any distribution, which does not concentrate too much probability mass on a single element - without this constraint the adversary can concentrate all of the mass on a single element and we end up with worst case analysis. Furthermore, they also assume that the random variables for edge weights are independent.
>
> Our approach is similar to the above, however, our constraints are even weaker. We only require a single element $e^*$ to not be chosen completely adversarially and will allow for partial dependence between the random variables.
>
> Questions
>
> 1) See above. Please let us know if you would like a more detailed comparison. We will also add this comparison to the final version of the paper.
>
> 2) Mini-batch methods (e.g., mini-batch k-means, SGD) are highly effective in practice and the standard choice for massive datasets. We empirically observe the same for submodular maximization, and justify these observations via smoothed analysis, thereby connecting theory and practice. Our analysis is simple and does not rely on any heavy machinery.

---

### Official Review · Reviewer_yRVE · 2025-10-31

**Soundness:** 3
**Presentation:** 3
**Contribution:** 2
**Rating:** 4
**Confidence:** 4

**Summary:**

This paper introduces a highly efficient mini-batch greedy algorithm for maximizing large-scale decomposable submodular functions ($F=\sum_{i=1}^{N}f^{i}$), where evaluating the full function is prohibitively expensive. To theoretically justify the empirical superiority of a simple uniform sampling strategy over complex weighted methods, the authors move beyond worst-case analysis and employ a novel smoothed analysis framework. This framework allows them to prove that their uniform mini-batch algorithm achieves near-optimal approximation guarantees for both cardinality and p-system constraints, with improved query complexity. Specifically, for a cardinality constraint, the algorithm achieves a $(1-1/e-\epsilon)$-approximation using only $\tilde{O}(\frac{k^{2}n}{\epsilon^{2}\phi})$ oracle queries. Extensive experiments on five real-world datasets substantiate these findings, demonstrating that the proposed method is orders of magnitude faster than the state-of-the-art while providing superior or comparable solution quality. The paper also empirically validates that the core assumptions of its theoretical model hold in these practical settings.

**Strengths:**

The paper is well-written and easy to follow. A major strength lies in its comprehensiveness, as the authors rigorously validate their approach across three distinct settings: cardinality constraints, p-systems, and curvature. Moreover, the emphasis on mini-batching represents a timely and meaningful contribution to the machine learning community, addressing scalability in submodular optimization.

**Weaknesses:**

The primary limitation stems from an overly strong assumption. Specifically, the authors assume that for any individual element $e$, each function $f_i$ satisfies both $f_i(e) \in [0,1]$ and an expected value $E[f_i(e)] \ge \phi$. These strict upper and lower bounds on single-element contributions overly simplify the problem, making it plausible that a simple uniform sampling strategy could already achieve a comparable approximation of the overall objective function $F$ using standard concentration results such as the Chernoff bound.

**Questions:**

1.	If we remove the normalization condition while keeping all other conditions unchanged, does the conclusion still hold?
2.	Alternatively, can you prove that without these assumptions, it is impossible to optimize using fewer than ( Nnk ) queries in terms of order?
3.	It seems that the paper still contains some minor errors — for example, in proof of lemma 3.4, the Chernoff bound expressions are missing the $\exp()$ wrapper.
4.	The authors did not mention that their assumptions may oversimplify the theoretical analysis of the algorithm.

---

> ### Author Response · Authors · 2025-11-15
>
> Thank you for your comments and questions.
>
> Weaknesses
>
> We do not assume that **for every** $e\in E$ it holds that $\forall i, E[f^i(e)] \geq \phi$. We only assume that there **exist a single** $e^* $ such that $\forall i, E[f^i(e^*)] \geq \phi$. All other values can be chosen adversarially. The assumption that $f^i \in [0,1]$ is without loss of generality as we can simply normalize the functions by their largest value - it is only there to avoid introducing additional notation for the maximum value a function takes.
>
> Questions
>
> 1) See explanation above.
> 2) Without these assumptions we may set all $f^i\equiv 0$ except for a single $f^j$. This means that we must consider all functions and will require at least $N$ queries. Assuming you want the optimal approximation ratio of $1-1/e$ the best known algorithm is the simple greedy algorithm in which case the total running time is indeed $O(Nnk)$. If you are willing to accept a $1-1/e - \epsilon$ approximation this can be improved slightly to $O(Nn\log 1/\epsilon)$. But there is no way to get rid of the $N$ factor in the worst-case.
> 3) Thanks for pointing this out. We will make sure to take care of this in the updated version.
> 4) Indeed assuming that for every $e\in E$ it holds that $\forall i, E[f^i(e)] \geq \phi$ is an over simplification. **We do not assume this.**

---

### Official Review · Reviewer_VhBo · 2025-11-01

**Soundness:** 3
**Presentation:** 3
**Contribution:** 3
**Rating:** 6
**Confidence:** 3

**Summary:**

A decomposable submodual rmaximization, the sub-modular function $F$ is the sum of $N$ other sub-modular functions. The algorithms that make oracle calls to $F$ need to make $N$ oracle calls to the smaller/simpler submodular functions, thus the run-time has undesirable dependency on $N$.  To address this, prior work proposed sparsifier approach, that choses a subset of the smaller functions (and scales them) with certain probabilities. However the calucation of these probabilities takes $O(Nn)$ oracles, and this is treated as preprocessing step, in prior works.

The authors observe that uniform sampling probabilities work very well in practice and attempt to come with a theoretical explanation (though in the worst-case, uniform sampling is bad).  They resort to smoothed analysis, pioneered by Spielman and Tang, to address this.

**Strengths:**

1. As far as I know, this is the first instance of applying smoothed analysis for this problem.
2. It is known that many heuristic algorithms work well in practice, though their approximation guarantees are bad in the worst case. Thus, there is a need to go beyond the worst-case analysis, and this work is a step in that direction.
3. Generally, when the exact oracles are replaced with approximate oracles that have a multiplicative error, the proofs go through, with multiplicative errors creeping into the approximation factors. This work relies on using additive-approximate oracles, where the additive approximation errors are based on the optimal values. They show that this type of approximation still yields good guarantees, which in my view is clever.
4. Once the framework is set, the proofs are not hard to follow, which is a strength in my view.

**Weaknesses:**

My biggest concern is the definition of the smoothing model. Where is the randomness coming from? The functions $f^i$ are all deterministic. So what does it mean to say $f^(e^*)$ is a random variable and so what does the Expectation of this reandom variable mean? Where is the underlying distribution? If I were to infer, my guess is that each $f^i$ is chosen with a certain probability, and that's where the randomness is coming from, but I am not sure. This must be addressed.

I am willing to increase my score once I understand this.

**Questions:**

1. Please see the weakness.
2. Definition of Smoothing model: Can you formally define "each RV depends on at most d others".
3. Similarly, what does bounded dependency mean in Theorem 3.2?

---

> ### Author Response · Authors · 2025-11-15
>
> Thank you for your comments and questions.
>
> **Definition of smoothing model**
>
> This is a great point. Let us contrast the smoothing model with the worst-case model.
>
> Worst case model: an adversary can choose all $f^i$ in an adversarial way. There is no constraint on how the functions are chosen.
>
> Our model: an adversary can choose a *distribution* from which every $f^i$ is sampled under some constraints. Note, if we do not add any constraints here then this is just equivalent to the worst case model. The adversary can choose a distribution where all of the density is concentrated on a single element. Our constraints force the adversary to choose non-pathological distributions. Formally, the adversary can choose any distribution for $f^i$ as long as the distributions have the two properties mentioned in our smoothing model.
>
> Crucially, each sampled $f^i$ is itself a deterministic function; the randomness comes from which deterministic function is drawn from the adversary-chosen distribution. Thus $f^i$ is a random variable with respect to this sampling process. The underlying probability space is therefore the product space generated by these adversarially chosen, but constrained, distributions.
>
>
> Models in the literature: Let us contrast our model with a commonly used model in the literature to highlight the generality of our model. We consider the widely studied Local Max-Cut problem (https://www.roeglin.org/publications/SODA14.pdf). Here the input is an edge weighted graph to which we want to apply smoothing. The standard smoothing model assumes that every edge weight is sampled from some distribution where the only constraint is that its density is upper bounded by $\phi$. Intuitively, this means that the adversary can choose any distribution, which does not concentrate too much probability mass on a single element - without this constraint the adversary can concentrate all of the mass on a single element and we end up with worst case analysis. Furthermore, they also assume that the random variables for edge weights are independent.
>
> Our approach is similar to the above, however, our constraints are even weaker. We only require a single element $e^*$ to not be chosen completely adversarially and will allow for partial dependence between the random variables.
>
> **Definition of bounded dependency**
>
> Consider a set of random variables $X_1,\dots,X_N$. For every pair of variables, they can be either independent (i.e.,
> $\Pr[X_i = a \wedge X_j = b] = \Pr[X_i = a] \Pr[X_j = b]$ for every $a,b$) or dependent.
> We construct a *dependency graph* as follows: the nodes correspond to the random variables, and we place an edge between two nodes $i$ and $j$ if $X_i$ and $X_j$ are dependent.
>
> Using this graph, we say that a random variable $X_i$ *depends on at most $d$ others* if its degree in the dependency graph is at most $d$.
> We say that the collection $\{X_1,\dots,X_N\}$ has *bounded dependency* with parameter $d$ if the maximum degree in the dependency graph is at most $d$.

---

### Official Review · Reviewer_9gZU · 2025-11-03

**Soundness:** 3
**Presentation:** 2
**Contribution:** 2
**Rating:** 2
**Confidence:** 4

**Summary:**

This work studies a uniform sampling algorithm for maximizing a nonnegative monotone *decomposable* subomdular function
$F = \sum_{i=1}^N f^i$, where each $f^i$ is a nonnegative monotone submodular function.
Specifically, it proposes a uniform sampling method to sparsify $F$ when $N$ is large by
sampling a subset of function $f^i$ of size $N' << N$.
They offer a smoothed analysis inspired by [Spielman--Teng, JACM 2004] to go beyond worst-case inputs.
Much of this work is focused on removing a $O(N n)$-time bottleneck step in the
sparsification scheme of [Kenneth--Krauthgamer, ICALP 2024] that computes the
values $p_i = \max_{e \in E, F(e)\ne 0} f^i(e) / F(e)$.
Finally, the authors give an empirical study across a wide range of datasets.

**Strengths:**

- Builds on sparsification framework for decomposable submodular functions in [Rafiey--Yoshida, AAAI 2022].
- Identifies a hard/pathological case where only a single $f^i$ takes nonzero
  values, and proposes smoothed analysis to bridge this gap (beyond worst-case analysis)
- Diverse set of experiments (e.g., several different interesting datasets, tasks, and objective functions).

**Weaknesses:**

- The writing is not as crisp as it could be: there are many technical ideas discussed that distract from the main contribution (e.g., the "Approximate oracles" paragraph in Section 2.1 doesn't add to the message and distracts the reader). The paper could benefit from presenting fewer ideas and making things more streamlined, without compromising the main message.
- The theoretical contribution is novel but limited (Section 3)

**Questions:**

**Questions**
- [049] What are practical examples of $N$ being extremely large? What are
  reasonable assumptions for $f^i$ to all be different? If they are the same
  function (e.g., functions for two students with the same preference), then
  this is the same as increasing the coefficient of $f^i$ and reducing $N$.

**Typos/suggestions**
- [120] Suggestion: The paper could benefit from presenting a more compelling
  running example than lunch menu optimization.
- [528] Typo: Update the Kenneth--Krauthgamer reference to ICALP 2024.

---

> ### Author Response · Authors · 2025-11-15
>
> Thank you for your comments and questions.
>
> Weaknesses
>
> 1) We will be happy to update the paper according to your suggestions. We can move the approximate oracle paragraph to the appendix. Any other changes you would like us to make?
>
> 2) Mini-batch methods (e.g., mini-batch k-means, SGD) are highly effective in practice and the standard choice for massive datasets. We empirically observe the same for submodular maximization, and justify these observations via smoothed analysis, thereby connecting theory and practice.
>
> Questions
>
> 1) Our experiments section contains real datasets where N is large. Several examples: Exemplar-based clustering - finding a small set of images that represent a large set of images (here N is the number of images in the large set). Influence Maximization - finding a subset of nodes that maximize the influence spread in the network (here N is the number of nodes in the network).
>
> 2) See the examples above. Furthermore, $f^i$ need not all be different. The pathological case is where nearly all are the same, which clearly cannot be the case in the real world examples presented above.

---

### Meta-Review · Area_Chair_XEwu · 2025-12-08

**Summary:**

The reviewers see potential in this work, but overall there appear to be a few too many doubts around issues including novelty, placement among prior work, and presentation.  At the very least, a major revision is likely to help address these points.  The response regarding [1] is useful but at least two others were mentioned, and there are likely more.  In addition, due to consistent confusion around the smoothing assumptions, they could be discussed better, perhaps with illustrative examples, and a clearer exposition of limitations / cases where it fails (it is arguably not as mild as indicated).

**Reviewer Concerns:**

I acknowledge that not all the reviewer concerns are correct; in particular, one reviewer confused "for all e" with "there exists e".  On the other hand, some of the most significant concerns would very likely still remain after the rebuttal, perhaps most notably technical novelty.  More examples are mentioned above.

**Reviewer Scores:**

My best judgment is that a proper discussion period would have at most led to mild score increases, with 2-3 reviewers still leaning towards rejection, in particular likely having reservations on the technical novelty.

---

### Decision · Program_Chairs · 2026-01-26

Reject